# Four ppm measurement of the antihydrogen ground-state hyperfine splitting

R. Akbari[1], L. O. de Araujo Azevedo[2], C. J. Baker[3], W. Bertsche[4,5], N. M. Bhatt[3,6], G. Bonomi[7,8], A. Capra[6], I. Carli[6], C. L. Cesar[2], M. Charlton[3], A. Cridland Mathad[3,9], A. Del Vincio[7,8,10], D. Duque Quiceno[1,6], S. Eriksson[3], A. Evans[1,6], J. Fajans[11,25], T. Friesen[12 ✉], M. C. Fujiwara[6,13], L. M. Golino[3], M. B. Gomes Gonçalves[3], J. S. Hangst[14 ✉], M. E. Hayden[15], P. Heidari[12], D. Hodgkinson[11], C. A. Isaac[3], S. A. Jones[16], S. Jonsell[17], N. Madsen[3], V. R. Marshall[14], J. T. K. McKenna[4], T. Momose[1,6,18], J. Nauta[9], A. N. Oliveira[4], A. Powell[9,12], C. Ø. Rasmussen[9,19], T. Robertson-Brown[3], F. Robicheaux[20], R. L. Sacramento[2], E. Sarid[21,22], J. Schoonwater[3], D. M. Silveira[2], J. Singh[4], G. Smith[1,6], C. So[6], S. Stracka[23], J. Suh[12], A. G. Swadling[12], T. D. Tharp[24], K. A. Thompson[3], R. I. Thompson[6,12], E. Thorpe-Woods[3], A. J. Uribe Jimenez[6,12 ✉], M. Urioni[7,8], D. P. van de Werf[3], S. G. Wilson[12], P. Woosaree[12], J. S. Wurtele[11] & The ALPHA Collaboration*

The hydrogen atom is a touchstone for the foundations, evolution and frontiers of quantum theory[1–9]. Key spectral lines of this atom have been determined to remarkable precision[10,11]. Our research focuses on the study of antihydrogen, the antimatter counterpart of hydrogen. We test fundamental symmetries of nature (such as simultaneous charge conjugation, parity inversion, and time reversal or CPT symmetry) through precision comparisons of these atomic systems[12]. Recent 1S–2S spectroscopic measurements on trapped antihydrogen have achieved relative precisions of parts per trillion (refs. 13,14). However, the ground-state hyperfine splitting, which is sensitive to the internal structure of the antiproton, has only been measured to 400 parts per million (ppm). Here we report a 4 ppm measurement of the antihydrogen ground-state hyperfine splitting energy $a_{1S}$, advancing the state-of-the-art precision[15] by two orders of magnitude. From microwave spectroscopy experiments with roughly 24,000 anti-atoms, we determine $a_{1S}/h = 1,420,404.8 \pm 1.1(\text{stat.}) \pm 5.6 \text{ (sys.)}$ kHz in a 1-T magnetic field, consistent with expectations for hydrogen[11]. At this level, our measurement is sensitive to the internal structure of the antiproton, which contributes at about 40 ppm and is approaching the limit of existing theoretical analyses[16]. The gains we report are the product of marked advances in magnetic trap field control, stabilization and characterization; anti-atom spin-state manipulation; and improved antihydrogen accumulation rate[17].

The observation of hyperfine structure in atomic hydrogen and the high-precision measurements of its zero-field ground-state splitting, at the level of seven parts in $10^{13}$ (1 mHz absolute uncertainty)[11,18,19], were landmark achievements. These results provided the first evidence of the anomalous magnetic moment and advanced quantum electrodynamics of the electron. In antihydrogen, measurement of the ground-state hyperfine splitting offers a powerful test of CPT symmetry, complementary to comparison of the 1S–2S transition frequencies measured with Doppler-free two-photon spectroscopy[20]. The hyperfine splitting frequency is a particularly sensitive probe of the internal structure of the antiproton through the Zemach correction and nuclear polarizability at the level of 40 parts per million (ppm) (refs. 16,21). Recently, the 2S hyperfine splitting of antihydrogen was inferred by combining laser spectroscopy of the 1S–2S transition with our previous determination of the ground-state hyperfine splitting[14]. However, the precision was strongly limited by the latter. Finally, precision measurements of the 2S and 1S hyperfine splittings can be combined to determine an experimental value for the Sternheim interval[22] in

[1]Department of Physics and Astronomy, University of British Columbia, Vancouver, British Columbia, Canada. [2]Instituto de Física, Universidade Federal do Rio de Janeiro, Rio de Janeiro, Brazil. [3]Department of Physics, Faculty of Science and Engineering, Swansea University, Swansea, UK. [4]School of Physics and Astronomy, University of Manchester, Manchester, UK. [5]Cockcroft Institute, Sci-Tech Daresbury, Warrington, UK. [6]TRIUMF, Vancouver, British Columbia, Canada. [7]University of Brescia, Brescia, Italy. [8]INFN Sezione di Pavia, Pavia, Italy. [9]Experimental Physics Department, CERN, Geneva, Switzerland. [10]University of Trento, Trento, Italy. [11]Department of Physics, University of California at Berkeley, Berkeley, CA, USA. [12]Department of Physics and Astronomy, University of Calgary, Calgary, Alberta, Canada. [13]WPI-QUP, KEK, Tsukuba, Japan. [14]Department of Physics and Astronomy, Aarhus University, Aarhus, Denmark. [15]Department of Physics, Simon Fraser University, Burnaby, British Columbia, Canada. [16]Van Swinderen Institute for Particle Physics and Gravity, University of Groningen, Groningen, The Netherlands. [17]Department of Physics, Stockholm University, Stockholm, Sweden. [18]Department of Chemistry, University of British Columbia, Vancouver, British Columbia, Canada. [19]Physics Department, Brookhaven National Laboratory, Upton, NY, USA. [20]Department of Physics and Astronomy, Purdue University, West Lafayette, IN, USA. [21]Ben Gurion University of the Negev, Be'er Sheva, Israel. [22]Soreq NRC, Yavne, Israel. [23]INFN Sezione di Pisa, Pisa, Italy. [24]Physics Department, Marquette University, Milwaukee, WI, USA. [25]Deceased: J. Fajans. *A list of authors appears at the end of the paper. ✉e-mail: timothy.friesen@ucalgary.ca; jeffrey.hangst@cern.ch; alberto.jesus.uribe.jimenez@cern.ch

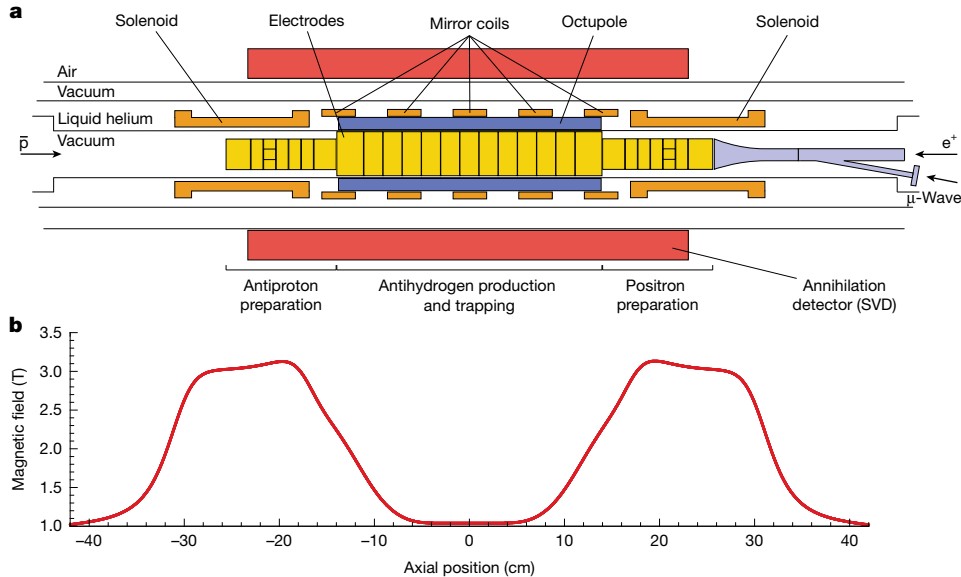

**Fig. 1 | Antihydrogen trapping apparatus and magnetic field. a**, Cross-sectional schematic of the ALPHA-2 apparatus, including magnets, Penning–Malmberg trap electrodes, annihilation detector and microwave injection system (μ-waves). Antiprotons ($\bar{\text{p}}$) and positrons ($e^+$) are injected from opposite ends of the apparatus, and the two solenoids in the preparation areas aid in cooling and compression of the charged particles. The 1 T external solenoid surrounding the trap is not shown. **b**, Calculated magnetic field profile along the trap axis.

antihydrogen, $(8a_{2S} - a_{1S})/h$, which is largely insensitive to nuclear-structure effects and provides a means to probe high-order quantum electrodynamic (QED) effects[16].

## Antihydrogen production and trapping

Our experiments are conducted at the CERN Antiproton Decelerator facility using the previously described ALPHA-2 antihydrogen apparatus[23]. Key elements of this device are shown in Fig. 1a. Antiproton ($\bar{\text{p}}$) and positron ($e^+$) plasmas held in a cylindrical Penning–Malmberg trap[24] are cooled, radially compressed and merged to synthesize antihydrogen. The latter occurs in a 0.54-K-deep magnetic potential well, in which (neutral) anti-atoms in appropriate spin states and with sufficiently low kinetic energy are confined; that is, they are prevented from interacting with the (matter) surfaces of the apparatus and annihilating. A typical antihydrogen production sequence lasts about 4 min and yields about 100 trapped anti-atoms. This number represents a marked improvement over previous antihydrogen measurements and results from the substantial decrease in positron plasma temperatures achieved by sympathetically cooling the $e^+$ with laser-cooled beryllium ions[17]. This production process can be repeated many times without releasing anti-atoms[25]. In this way, we accumulate samples of roughly 1,500 atoms from consecutive cycles over approximately 1 h.

The on-axis magnetic trapping potential for anti-atoms is shown in Fig. 1b. It is formed by the superposition of magnetic fields produced by currents in several superconducting coils in a common cryostat: an octupole, five short solenoids or 'mirror coils' and two longer, end solenoids. An external superconducting solenoid surrounds the atom trap and provides a uniform 1 T field for charged-particle confinement. The octupole provides radial $\bar{H}$ confinement; the two outermost mirror coils provide axial $\bar{H}$ confinement; and the end solenoids aid in cooling and compression of charged particles. The three remaining mirror coils are used to shape the magnetic field for spectroscopy (see below). Currents in the octupole, solenoids and mirror coils are actively controlled and stabilized using feedback from direct current current transformers (DCCTs). The current in the external solenoid is set to a target value, and the magnet is then disconnected from its power supply while current continues to flow in the superconductor (persistent current mode). Microwaves are injected into the apparatus through a custom-built vacuum window and propagate down a waveguide to the trapping region. Antihydrogen annihilation events are monitored by tracking the trajectories of antiproton annihilation products (charged pions) through a three-layer silicon vertex detector (SVD) with approximately 3π solid-angle coverage of the trap midpoint[23,26] (Methods).

## The spectroscopy experiment

The positronic ground state of antihydrogen is split into four hyperfine sublevels that we denote as $|a\rangle$, $|b\rangle$, $|c\rangle$ and $|d\rangle$, in order of increasing energy in weak magnetic fields (Fig. 2a). Anti-atoms in states $|c\rangle$ and $|d\rangle$ are low-field seekers and can be trapped in the vicinity of a magnetic minimum; anti-atoms in states $|a\rangle$ and $|b\rangle$ are high-field seekers and are rapidly ejected from our trap and annihilate on the surrounding surfaces. The current experiment uses microwaves at frequencies between 28 GHz and 31 GHz to resonantly and sequentially drive the positron spin-flip transitions $|c\rangle \to |b\rangle$ and $|d\rangle \to |a\rangle$. Near 1 T, the transition frequencies depend approximately linearly on the magnetic field strength with a slope of 28 GHz T$^{-1}$. The highly inhomogeneous magnetic field of our trap, with a depth of 0.8 T (set by the radial confining field), results in a strong positional dependence of the resonant frequencies. We therefore tailor the field to be as flat as possible, while ensuring the existence of a shallow centrally located axial field minimum (Methods and Extended Data Fig. 1) and focus on inducing transitions at the minimum of the magnetic potential. First, the $|c\rangle \to |b\rangle$ transition is driven at a monotonically increasing frequency, $f_{cb}$, in the form of a staircase function, until all $|c\rangle$-state anti-atoms are nominally eliminated, at which point the $|d\rangle \to |a\rangle$ transition is driven in the same manner at a frequency $f_{da} > f_{cb}$ (Fig. 2b). For each transition, we look for the onset of annihilation events as anti-atoms traversing the absolute minimum in the magnetic potential are brought into resonance. To the extent that the minimum magnetic field is the same for both transitions, and assuming the Breit–Rabi diagram for ground-state antihydrogen is analogous in structure to that of hydrogen, the corresponding frequency difference $f_{da}(B_{min}) - f_{cb}(B_{min})$ is equal to the antihydrogen hyperfine splitting frequency $a_{1S}/h$ (Methods). Note that the order in which transitions are driven cannot be reversed (that is, $|d\rangle \to |a\rangle$ followed by $|c\rangle \to |b\rangle$) because $|c\rangle$-state atoms at high magnetic fields

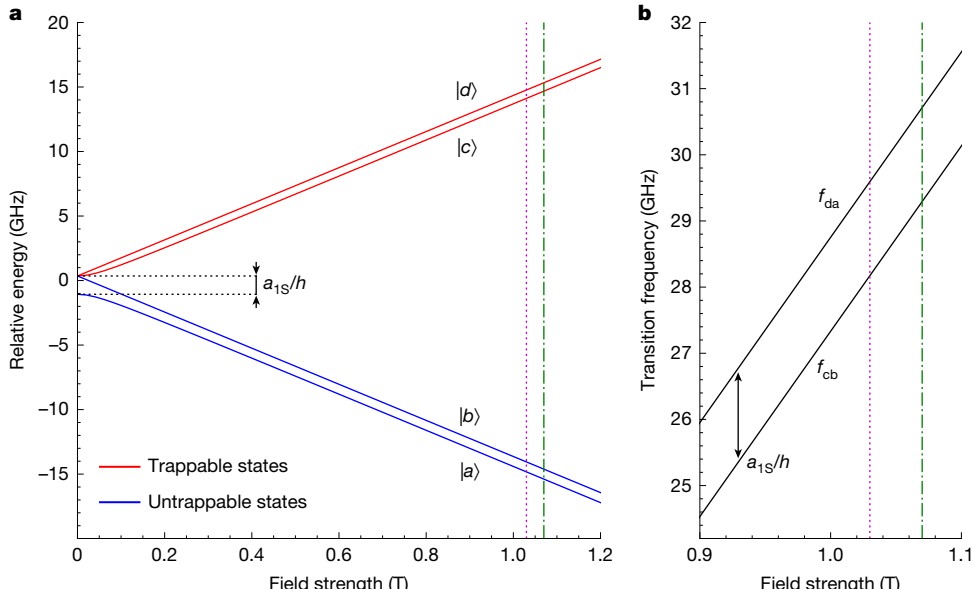

**Fig. 2 | Ground-state energy-level diagram and transition frequencies.**
**a**, Relative energies of the four hyperfine sublevels of antihydrogen's ground state as a function of magnetic field strength. The dotted purple line and dot-dashed green line indicate the axial magnetic minimum fields (1.03 T and 1.07 T), in which the two spectroscopy experiments were performed. **b**, Frequencies of the two positron spin-flip transitions used for spectroscopy $|d\rangle \leftrightarrow |a\rangle$ and $|c\rangle \leftrightarrow |b\rangle$ as a function of magnetic field strength. Frequencies are calculated assuming the properties of antihydrogen mirror those of hydrogen.

will come into resonance when driving $|d\rangle \rightarrow |a\rangle$ transitions near the magnetic minimum. This will remove $|c\rangle$-state atoms from the trap and create a high annihilation background that will mask the onset of $|d\rangle$-state atom annihilations. Similarly, we must scan from low to high frequencies for each transition to obtain a clear annihilation signal when the resonances at the magnetic minimum are reached.

We performed two hyperfine splitting experiments at two different base magnetic fields by changing the current in the external solenoid. In both experiments, the magnetic trap shape is maintained, but the first has an on-axis magnetic minimum of 1.03 T and the second of 1.07 T, which corresponds to a 1.1 GHz shift in transition frequencies. In each experiment, the external solenoid is set to the target current and then put into persistent current mode. We then energize the trap magnets to a fixed and stabilized current to create the flattened magnetic trap for the remainder of the experiment. Following this ramp, we observe the minimum magnetic field strength initially rise, level off and then enter a linear decay region, reducing the frequency of the positron spin-flip transitions by roughly 74 kHz per hour for the remainder of the experiment (Methods). We attribute these drifts to fine-scale flux

redistribution in the superconductors of the trap magnets, combined with the decay of the persistent current of the external solenoid. After the trapping magnets are energized, we wait a minimum of 1.5 h before beginning spectroscopic experiments to ensure the magnetic field evolution is in the linear decay region.

Both hyperfine splitting experiments consist of eight replicates that characterize the $|c\rangle \rightarrow |b\rangle$ and $|d\rangle \rightarrow |a\rangle$ resonances at the magnetic minimum. Each replicate begins with the accumulation of roughly 1,500 antihydrogen atoms from repeated positron–antiproton mixing cycles executed over a period of approximately 1 h. After each mixing cycle, pulsed electric fields are applied to eject any remaining charged particles. Microwaves are then introduced in four phases. Phase 1 involves a 48-step frequency staircase, with a step size of $\Delta f = 5$ kHz and a step duration of $T_s = 8$ s. We start Phase 1 roughly 100 kHz below the expected $|c\rangle \rightarrow |b\rangle$ resonance and scan 240 kHz to characterize the resonance at the magnetic minimum. After Phase 1 we observe that some $|c\rangle$-state anti-atoms remain in the trap, which we attribute to anti-atoms on orbits that did not come into resonance near the magnetic minimum. Left in the trap, these anti-atoms would present an increased annihilation

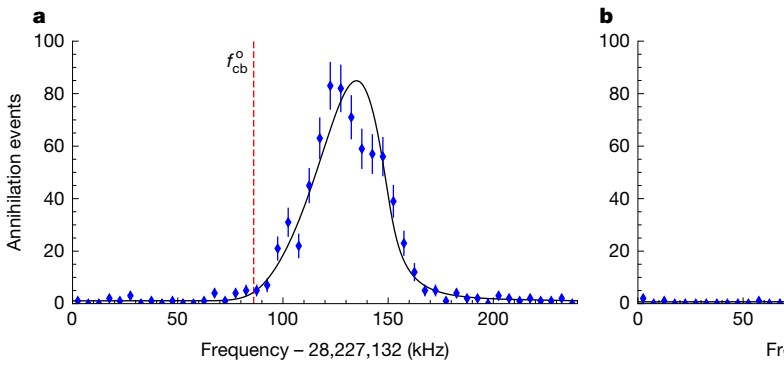

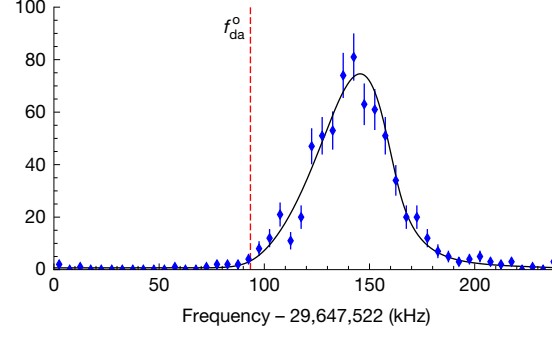

**Fig. 3 | Example positron spin resonance lineshapes. a**, An example of the $|c\rangle \rightarrow |b\rangle$ resonance lineshape obtained by plotting annihilation events against microwave frequency during Phase 1 of a replicate from the 1.03 T experiment. **b**, The $|d\rangle \rightarrow |a\rangle$ resonance during Phase 3 of the same replicate. The black lines are the empirical lineshape model, and the vertical red dashed lines indicate the onset frequency parameters, $f_{cb}^o$ and $f_{da}^o$. The error bars represent 1 standard deviation of counting errors.

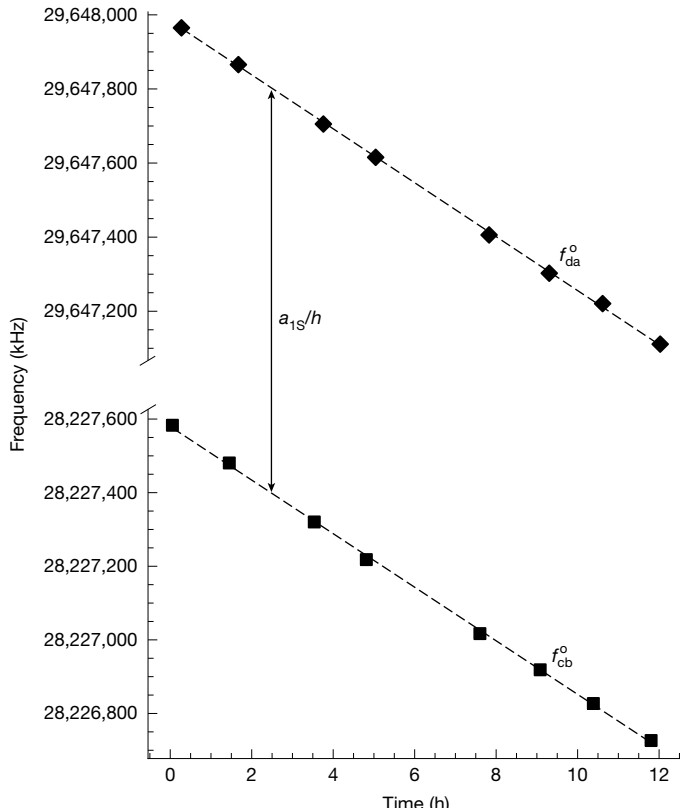

**Fig. 4 | Data and best-fit model.** Pairs of onset frequencies are determined from each replicate of the experiment of the 1.03 T experiment. Squares and diamonds denote $f_{cb}^{o}$ and $f_{da}^{o}$, respectively, found by fitting the model to the measured resonance lineshape of each replicate. The error bars are not visible on this scale. Dashed lines represent the best-fit linear decay model.

background when characterizing the $|d\rangle \rightarrow |a\rangle$ resonance. Phase 2 removes these $|c\rangle$-state anti-atoms with a 16-step frequency staircase, with the same step size and duration as Phase 1 but roughly 1 MHz below the expected $|d\rangle \rightarrow |a\rangle$ resonance. This higher frequency range increases the size of the resonant volume and is observed to effectively remove the remaining $|c\rangle$-state anti-atoms. The microwave field slightly heats the trap electrodes so following Phase 2 the microwave drive is switched off for 300 s to allow them to cool. Heating of the trap electrodes will increase the rate of outgassing and could result in a higher rate of antihydrogen annihilations on background gas during phases 3 and 4. Phase 3 is identical to Phase 1 except that the frequency at every step is higher by 1,420,390 kHz (corresponding to the hyperfine splitting for hydrogen minus the expected magnetic field drift), thus probing the $|d\rangle \rightarrow |a\rangle$ transition. Finally, Phase 4 removes the remaining $|d\rangle$-state anti-atoms by injecting frequencies high above the minimum $|d\rangle \rightarrow |a\rangle$ resonance. This ensures that there are no residual $|d\rangle$-state anti-atoms in the trap when the next replicate begins.

We track the magnetic field drift by systematically decrementing the frequency at which each subsequent frequency staircase begins, such that the atoms are ejected in the middle of the spectroscopy windows. Owing to the complex electromagnetic environment, the spatial microwave field structure inside the trap is not well understood and changes with frequency; hence, we choose the injected microwave powers in phases 1 and 3 to balance the positron spin-flip rates observed in a set of auxiliary experiments (Methods). In each experiment, the overall number of annihilations from ejected $|c\rangle$-state atoms was consistent with the number of events from ejected $|d\rangle$-state atoms. This indicates that equal numbers of $|c\rangle$-state and $|d\rangle$-state are produced and trapped and that the positron spin-flip rates for each transition are roughly equal.

## Analysis and results

The annihilation of ejected $\overline{H}$ detected by the SVD during Phase 1 and Phase 3 of each replicate yields resonance lineshapes for the $|c\rangle \rightarrow |b\rangle$ and $|d\rangle \rightarrow |a\rangle$ transitions, respectively (Fig. 3). The lineshapes reflect contributions from the structure of the non-uniform magnetic field, the depletion of the antihydrogen population as the microwave frequency is incremented, the orbits of the atoms through the magnetic field, the local amplitude of the microwave magnetic field seen by the atoms, and motional broadening effects. Without full knowledge of these factors, we cannot develop a precise physical model of the observed lineshapes. Instead, we take the approach of fitting the lineshapes with an empirical model with a frequency parameter ($f_{cb}^{o}, f_{da}^{o}$), associated with the onset of annihilation events as anti-atoms traversing the magnetic minimum are brought into resonance (Methods). These parameters are proxies for $f_{cb}$ ($B_{min}$) and $f_{da}$ ($B_{min}$), respectively, and share the same property that $a_{1S}/h = f_{da}^{o} - f_{cb}^{o}$ if measured at the same magnetic field. Alternative frequency parameters and models can also be used to extract $a_{1S}/h$. For example, consistent results are obtained using the peak ($f_{max}$) of the empirical lineshapes (Methods) rather than the onset. Because we are performing a difference measurement and not extracting absolute frequencies for the individual transitions, systematic effects associated with our choice of model largely cancel and the residual systematics can be quantified (Methods).

Because the minimum magnetic field magnitude is decreasing in time, it is different between Phase 1 and Phase 3 (which are separated by 812 s) of each replicate as well as between replicates. To account for this, we assume the magnetic field strength decay is purely linear and fit the two sets of onset parameters with two straight lines with a common slope as shown in Fig. 4. The other two fit parameters are the average of the two intercepts and the separation between the two lines, the latter of which is our estimation of $a_{1S}/h$.

The results for the hyperfine splitting frequencies inferred for the two datasets, collected on separate days at the base magnetic fields of 1.03 T and 1.07 T, are summarized in Table 1, along with statistical and systematic uncertainties (Methods). The latter account for the choice of the empirical function and of the fixed parameter values in the fit (signal model), the possible variation of the signal distributions (for example, due to changes in the microwave field structure) across replicates (reproducibility), potential deviations from a purely linear magnetic field decrease (B-drift), and the discrete nature of the frequency scan and mapping between annihilation time and frequency (binning). The larger reproducibility uncertainty in the 1.03 T experiment could be attributed to a larger variation in the microwave magnetic field strength seen by the anti-atoms as a function of frequency between replicates. We combine the statistically independent results of the measurements for the two datasets, taking into account correlations between systematic uncertainties, to obtain $a_{1S}/h = 1,420,404.8 \pm 1.1$ (stat.) $\pm 5.6$ (sys.) kHz, consistent with CPT invariance.

**Table 1 | Results for the hyperfine splitting and the associated uncertainties for the two base magnetic field values**

| B-field (T) | $a_{1S}/h$ (kHz) | Statistical uncertainty (kHz) | Systematic uncertainty budget (kHz) | | | | |
|---|---|---|---|---|---|---|---|
| | | | Reproducibility | Signal model | Binning | B-drift | Total |
| 1.03 | 1,420,403.9 | 1.7 | 7.0 | 4.8 | 2.0 | 1.2 | 8.8 |
| 1.07 | 1,420,405.8 | 1.5 | 2.8 | 5.5 | 2.0 | 2.6 | 7.0 |

## Conclusions

The leap in precision accompanying our measurement exposes the regime in which quantum electrodynamics (QED) computations of $a_{1S}$ are challenged by nuclear-structure effects (dominated by the Zemach correction)[16], which are of interest in the context of the proton radius puzzle[9,27–29]. Moreover, synergy exists between the present investigation and parallel efforts to characterize two-photon optical transitions between the 1S and 2S manifolds of antihydrogen. Recently, the 2S hyperfine splitting, $a_{2S}/h$, was determined[14] to a few parts per thousand by combining our previous 1S hyperfine measurement[15] with new spectroscopic results of the 1S to 2S transition. However, the precision was limited by the uncertainty in $a_{1S}/h$. With this new measurement, we have also improved the precision of the $a_{2S}/h$ determination by a factor of 26 to obtain $a_{2S}/h = 177{,}563 \pm 18$ kHz, in agreement with the most accurate hydrogen measurements[30]. Furthermore, it establishes an experimental bound for the Sternheim interval in antihydrogen of $(8a_{2S} - a_{1S})/h = 100 \pm 150$ kHz (ref. 22), where the precision is now limited by the measurement of $a_{2S}/h$. The Sternheim interval cancels out leading nuclear-structure effects that affect both the ground-state hyperfine splitting and the 1S–2S transition frequency, thus providing a probe of higher-order QED contributions. This interval is predicted by QED theory in hydrogen to be $48.953 \pm 0.003$ kHz (ref. 31), and its most precise measurement is $48.9592 \pm 0.0068$ kHz (ref. 30). Although our current determination is still five orders of magnitude worse than that of hydrogen, future 1S–2S spectroscopy measurements to determine $a_{2S}/h$ in antihydrogen can improve this substantially.

Adaptations of methods introduced here will enable coordinated comparisons of the mean transition frequency with electron cyclotron resonance magnetometry[32] and thereby probe relativistic and radiative contributions to the bound-state magnetic moment of the anti-atom in a manner that is more direct than has been possible with hydrogen[33]. Future measurements will focus on the nuclear magnetic resonance transition ($|c\rangle \leftrightarrow |d\rangle$) at 0.65 T, where the transition frequency passes through a maximum[34]. Operating at this turning point suppresses sensitivity to major systematic effects connected to the non-uniform magnetic trap and magnetic field variations. In combination with recent advances in laser and adiabatic expansion cooling[35,36], we expect that this measurement will improve the ground-state hyperfine splitting determination by another two orders of magnitude. A measurement of $a_{1S}$ complementary to that reported here, using an antihydrogen beam, is being pursued by the ASACUSA Collaboration at CERN[37]. A recent, marked advance in manipulating the spin of trapped antiprotons[38] is a close relative of the current work, and promises increased precision in the measurement of the antiproton magnetic moment.

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

**The ALPHA Collaboration**

R. Akbari¹, L. O. de Araujo Azevedo², C. J. Baker³, W. Bertsche⁴,⁵, N. M. Bhatt³,⁶, G. Bonomi⁷,⁸, A. Capra⁶, I. Carli⁶, C. L. Cesar², M. Charlton³, A. Cridland Mathad³,⁹, A. Del Vincio⁷,⁸,¹⁰, D. Duque Quiceno¹,⁶, S. Eriksson³, A. Evans¹,⁶, J. Fajans¹¹,²⁵, T. Friesen¹², M. C. Fujiwara⁶,¹³, L. M. Golino³, M. B. Gomes Gonçalves³, J. S. Hangst¹⁴, M. E. Hayden¹⁵, P. Heidari¹², D. Hodgkinson¹¹, C. A. Isaac³, S. A. Jones¹⁶, S. Jonsell¹⁷, N. Madsen³, V. R. Marshall¹⁴, J. T. K. McKenna⁴, T. Momose¹,⁶,¹⁸, J. Nauta⁹, A. N. Oliveira⁴, A. Powell⁹,¹², C. Ø. Rasmussen⁹,¹⁹, T. Robertson-Brown³, F. Robicheaux²⁰, R. L. Sacramento², E. Sarid²¹,²², J. Schoonwater³, D. M. Silveira², J. Singh⁴, G. Smith¹,⁶, C. So⁶, S. Stracka²³, J. Suh¹², A. G. Swadling¹², T. D. Tharp²⁴, K. A. Thompson³, R. I. Thompson⁶,¹², E. Thorpe-Woods³, A. J. Uribe Jimenez⁶,¹², M. Urioni⁷,⁸, D. P. van der Werf³, S. G. Wilson¹², P. Woosaree¹² & J. S. Wurtele¹¹

# Methods

## Trap magnetic field profile, stabilization and characterization

The design of the flattened magnetic field profile along the axis at the centre of the trap in the trapping region, shown in Fig. 1, represents an important component of the gains in frequency resolution reported here. In particular, the curvature of the magnetic field governs the time-scale over which anti-atoms interact with the microwave field as they are brought into resonance. In round numbers, the second axial derivative of the axial magnetic field profile used in the current work is less than $2 \text{ T m}^{-2}$, or a factor of 20 smaller than that of the (unflattened) trapping field used during the measurement of the hyperfine splitting reported in ref. 15. This, in turn, enhances positron spin-flip efficiencies and our ability to probe anti-atoms much closer to the minimum frequencies of the two transitions. However, unlike the maximally flattened field configurations used in previous optical spectroscopy experiments[13,39], here we intentionally depress the field below the central mirror coil by about $5 \times 10^{-5}$ T to ensure the existence of a shallow, centrally located absolute minimum (Extended Data Fig. 1).

We use the electron cyclotron resonance (ECR) technique to map the axial magnetic field profile along the axis of the trap in situ with a frequency resolution of the order of 1 ppm (ref. 32) and a spatial resolution of about 1 mm, and are thus able to directly confirm the formation, location and depth of the central magnetic field minimum along the axis. Moreover, we use ECR to independently resolve and characterize both the initial field drift (attributed to fine-scale flux relaxation or redistribution after the first ramp up of the superconducting magnets at the beginning of the experiment) and the subsequent long-term downward linear field drift (attributed to the decay of persistent currents in the external solenoid used to generate the 1 T background field). These studies show that the drifts depend on the time history of magnet operations. Consequently, magnets are energized in the same sequence in all experiments.

For each experiment, we perform ECR measurements after the magnetic trap is energized as well as after each replicate as a complementary monitor of the magnetic field drift. From a linear fit to the ECR magnetic field measurements following each replicate, we extract a linear magnetic field drift of $-0.025 \pm 0.001$ G h$^{-1}$ for the 1.03 T experiment and $-0.026 \pm 0.001$ G h$^{-1}$ for the 1.07 T experiment (Extended Data Fig. 2). Assuming the same magnetic field dependence as in hydrogen, this corresponds to expected positron spin resonance onset frequency drifts of $-71 \pm 3$ kHz h$^{-1}$ for the 1.03 T experiment and $-73 \pm 3$ kHz h$^{-1}$ for the 1.07 T experiment. This is consistent with the measured linear drift of the positron spin resonance frequencies of $-72.82 \pm 0.04$ kHz h$^{-1}$ for the 1.03 T experiment and $-75.64 \pm 0.05$ kHz h$^{-1}$ for the 1.07 T experiment we find in our analysis (Fig. 4). Note that ECR measurements of the on-axis magnetic field were used solely to characterize magnetic fields in preparation for the experiments described here and as a complementary magnetic field measurement. ECR measurements were not used in our extraction of the ground-state hyperfine splitting of antihydrogen.

During our experiments, the mirror coils are supplied with currents ranging from a few amperes to nearly 500 A. The octupole is operated with a current of 900 A. These currents are individually monitored using high-precision, ultrastable direct current–current transformers based on closed-loop fluxgate magnetometer sensors (ITZ 2000-SB FLEX ULTRASTAB from LEM International SA). The DCCT provides a ±10 V output with a 500 kHz small-signal bandwidth. This signal was digitized by 24-bit National Instruments NI-9239 cRIO (Compact Real-time Input Output) analog-to-digital converter modules at 50 kS s$^{-1}$ and averaged by the cRIO FPGA (field programmable gate array) firmware to 10 kS s$^{-1}$. The resulting averaged signal was used for proportional–integral–derivative (PID)-based closed-loop control of the magnet power supplies, stabilizing output currents to within several mA.

## Extracting the zero-field ground-state hyperfine splitting

The Breit–Rabi formula[40] gives the following energy levels of hydrogen (Fig. 2a) in a magnetic field:

$$E_d = \frac{a_{1S}}{4} + \frac{1}{2}g_e\mu_B B\left(1 - \frac{g_p m_e}{g_e m_p}\right),$$

$$E_c = -\frac{a_{1S}}{4} + \frac{1}{2}\left[a_{1S}^2 + \left(g_e\mu_B B\left(1 + \frac{g_p m_e}{g_e m_p}\right)\right)^2\right]^{1/2},$$

$$E_b = \frac{a_{1S}}{4} - \frac{1}{2}g_e\mu_B B\left(1 - \frac{g_p m_e}{g_e m_p}\right),$$

$$E_a = -\frac{a_{1S}}{4} - \frac{1}{2}\left[a_{1S}^2 + \left(g_e\mu_B B\left(1 + \frac{g_p m_e}{g_e m_p}\right)\right)^2\right]^{1/2},$$

where $\mu_B$ is the Bohr magneton, $g_e$ and $g_p$ are the electron and proton $g$-factors, and $m_e$ and $m_p$ are the electron and proton masses. By taking the difference of the transition frequencies, the magnetic field terms of the Breit–Rabi formula cancel and the hyperfine splitting $a_{1S}/h$ is extracted. Here, we assume that the ground-state energy levels of antihydrogen follow the same functional form as in hydrogen, and thus $a_{1S}/h = f_{da}(B) - f_{cb}(B)$ for any $B$, but the zero-field hyperfine splitting and magnetic field scaling coefficients may differ.

## Microwave magnetic field and positron spin-flip rates

Microwaves are produced using an Agilent E8257D PSG analog signal generator, amplified using a Miteq AMF-4B amplifier, and simultaneously matched to the ALPHA-2 apparatus in the vicinity of frequencies needed for spectroscopy using an E–H tuner. They enter the ultrahigh vacuum region of the apparatus through a custom-built vacuum window and propagate down a waveguide to the electrode stack, in which they enter the trapping volume, as shown in Fig. 1. A characterization of the microwave hardware was conducted using a Keysight model N5224B PNA Microwave Network Analyser. The signal generator output frequency is referenced to a 10 MHz signal provided by CERN, which has a precision better than $10^{-6}$ Hz. The signal generator calibration was verified before spectroscopy experiments were performed.

The positron spin-flip transitions are driven by the component of the microwave magnetic field that is transverse to the static axial magnetic field at the centre of the trap. Therefore, we are interested in balancing the amplitudes of this component of the microwave magnetic fields at frequencies in the vicinity of the $|c\rangle \to |b\rangle$ and $|d\rangle \to |a\rangle$ onset transition frequencies. We do this with ancillary microwave power studies, in which the lineshapes are compared for different injected powers using large samples of antihydrogen atoms (roughly 5,000). Injected powers (Extended Data Table 1) were chosen such that $|c\rangle \to |b\rangle$ and $|d\rangle \to |a\rangle$ lineshapes match as closely as possible. Differences in injected powers between phases 1 and 3 are the consequence of the strong frequency dependence of the power that reaches the anti-atoms. Any remaining imbalance of the positron spin-flip rates contributes to the overall systematic uncertainty (signal model in Table 1 and Methods, 'Systematic uncertainties'). During phases 2 and 4, the injected powers were increased to more quickly remove the remaining $|c\rangle$- and $|d\rangle$-state atoms, respectively. In both experiments, the number of annihilations attributed to $|c\rangle$-state atoms was consistent with the number attributed to $|d\rangle$-state atoms (Extended Data Table 1). This is consistent with the assumption that the two states are produced in equal amounts during formation and removed from the trap at consistent rates during the microwave sweeps.

## Data acquisition and selection

Antihydrogen annihilation events are reconstructed from their charged-particle products, which are detected by the SVD as described in previous experiments[13,41]. Annihilations are separated from the cosmic-ray background using machine-learning procedures based on a Boosted Decision Tree classifier[42]. The efficiency for annihilation event candidate selection is 75.7%, and the rate at which cosmic rays are misidentified as signal is $37.4 \times 10^{-3}$ s$^{-1}$.

Higher rates of background events as compared with the expected cosmic-ray contribution may result from factors ranging from antihydrogen annihilation on residual gas to microwave ejection of remnant trapped $|c\rangle$-state anti-atoms in Phase 3 as they come into resonance at higher magnetic fields. In our analysis, we make no attempt to disentangle different sources of background.

Annihilation events occurring during the frequency settling time (8–30 ms) between each microwave irradiation step are removed from the dataset. The remaining events are binned according to the frequency of the microwave radiation injected in the trap at the time of the annihilation.

## Data analysis

The distributions for each transition from each replicate of our experiment are fitted with an empirical model, defined by the convolution of a base lineshape function, $g$, and a resolution function, $R$, on top of a constant background, whose normalizations are free parameters of the fit. The base lineshape function is zero for frequencies below a given threshold, which we refer to as the onset frequency ($f^\circ$), and monotonically rises as the resonant volume increases and decays as the antihydrogen population is depleted. The resolution function accounts for broadening effects that smooth the lineshape, such as Doppler broadening, transit-time broadening and fluctuations in the magnetic field.

The $g$ function assumes the following form:

$$g(\overline{x}) \propto \begin{cases} 0 & \overline{x} < -\sigma(k+1), \\ \exp\left(-\frac{1}{2}\right) \times \left[\frac{\overline{x} + \sigma(k+1)}{\sigma k}\right]^k & [\overline{x} \geq -\sigma(k+1)] \wedge (\overline{x} < \sigma), \\ \exp\left(-\frac{\overline{x}^2}{2\sigma^2}\right) & (\overline{x} \geq -\sigma) \wedge (\overline{x} < 0), \\ \exp\left(-\frac{\overline{x}^2}{2\sigma_r^2}\right) & (\overline{x} \geq 0) \wedge (\overline{x} < \sigma_r), \\ \exp\left(-\frac{1}{2}\right) \times \left[\frac{\overline{x} + \sigma_r(n-1)}{\sigma_r n}\right]^{-n} & \overline{x} \geq \sigma_r. \end{cases}$$

Here, $\overline{x} = x - [f^\circ + \sigma(k+1)]$ and $f^\circ$ is the frequency threshold below which the piecewise function $g$ is zero and above which it features a power-law rise, corresponding to the rapid increase in resonant volume. The core of $g$ consists of an asymmetric Gaussian, followed by a power-law tail, which describes the region in which the annihilation counts decay due to the declining antihydrogen population for the state under study. The widths $\sigma$ and $\sigma_r$ of the asymmetric Gaussian core control the rise and decay lengths, respectively. The parameters $k$ and $n$ govern the power-law rise and decay, respectively.

Frequency sweeps proceed monotonically upwards, so our resolution function is asymmetric and defined as

$$R(\overline{x}) \propto \begin{cases} \exp\left(-\frac{\overline{x}^2}{2\sigma_b^2}\right) & \overline{x} \geq 0, \\ \exp\left(-\left|\frac{x}{\xi_R}\right|\right) & \overline{x} < 0, \end{cases}$$

where $\sigma_b$ is the broadening parameter and the right-sided exponential decay with a negligible width $\xi_R = 0.2$ kHz is included to ensure continuity.

For each magnetic field dataset, we perform one combined fit across all eight replicates for the $|c\rangle \to |b\rangle$ transition (Phase 1 annihilations) and another combined fit across the eight replicates for the $|d\rangle \to |a\rangle$ transitions (Phase 3 annihilations). The fit parameters controlling the lengths of the rising and falling edges of the empirical base function ($n$, $\sigma$, $\sigma_r$) are shared between all replicates but are allowed to differ between $|c\rangle \to |b\rangle$ and $|d\rangle \to |a\rangle$ transitions to accommodate residual differences in the respective spin-flip rates. For example, Fig. 3 shows a slightly larger lineshape width for the $|d\rangle \to |a\rangle$ transition compared with the $|c\rangle \to |b\rangle$ in the 1.03 T experiment. This indicates that the spin-flip rates at the $|d\rangle \to |a\rangle$ frequencies are slightly lower than at the $|c\rangle \to |b\rangle$ frequencies, despite our attempts to balance them in preparatory measurements (as described above). The parameters describing the steepness of the rising edges and the width of the resolution functions are fixed according to fits to simulated data (see below) of the experiment to be $k = 2$ and $\sigma_b = 10$ kHz, respectively, for both transitions. The uncertainties associated with these choices are discussed below. Only the onset frequencies, $f_{cb}^o$ and $f_{da}^o$, as well as the normalizations for the signal and background are allowed to vary for each repetition. The combined fits for each magnetic field dataset, therefore, yield eight pairs of onset frequency markers.

The magnetic field conditions for each of the replicates are different because the two transitions are probed at different times (separated by 812 s) while the magnetic field is drifting. To account for this effect, the two sets of onset frequencies are fit with a model function comprising two straight lines with a common slope whose separation in frequency is our estimate of $a_{1S}/h$, as shown in Fig. 4. The other fit parameter is the average of the two intercepts.

The distributions of the two histograms for a specific replicate are shown in Fig. 3. For each experiment (1.03 T and 1.07 T), a simultaneous maximum likelihood fit is performed to the eight replicates.

## Simulations

To inform two parameters of our model and study the associated systematic uncertainties we use simulations of the antihydrogen motion in the trap and the interaction with microwaves, reflecting the experimental frequency ramp and a magnetic field model tuned according to ECR measurements. Although these simulations cannot fully model the observed lineshapes at the required precision for this experiment because of the lack of knowledge of the microwave field and the static magnetic field off-axis, they offer a method to study the sensitivity of the lineshapes to different microwave powers and magnetic field profiles.

Simulations of antihydrogen motion in the ALPHA-2 trap were similar to those described in ref. 43. Positron spin resonance transitions are driven by the oscillating magnetic field component of the microwave field. Because we do not have knowledge of the position-dependent intensity of the microwave field, we use a position-independent intensity and polarization. To simulate positron spin resonance transitions, we performed a numerical, quantum two-state calculation in the resonance region. The detuning of each anti-atom was tracked in time and, when it was below 300 kHz, we calculated the quantum transition using the Crank–Nicolson algorithm with a time step of 39 ns. This quantum calculation uses the time-dependent magnetic field (hence a time-dependent detuning). After the atoms reach a detuning larger than 300 kHz, the probability of a transition is compared with a random number between 0 and 1. If the random number is less than the transition probability, then a spin flip was determined to have happened and the motion of the anti-atom is propagated using a flipped magnetic moment, pulling it towards the trap walls.

## Systematic uncertainties

The following sources of systematic uncertainties were identified and estimated. The overall uncertainty budget is provided in Table 1.

**Reproducibility.** In the analysis, the annihilation distribution for each transition is assumed to be the same for the full duration of one experiment. However, the annihilation distributions may vary across the eight replicates. This could be due to the injected microwave frequencies reducing by roughly 900 kHz over the course of each experiment, because of the decaying solenoid magnetic field, which could change the microwave magnetic field strength seen by the anti-atoms and modify the annihilation lineshapes. These variations have, therefore, been considered as a source of systematic uncertainty. To estimate this contribution, we repeated the fit procedure with shape parameters of $g(\bar{x})$ fixed using a fit to a high-statistics sample (consisting of roughly 5,000 anti-atom annihilations). These samples were collected following each experiment and had resonance frequencies of roughly 1,400 kHz and 1,200 kHz, respectively, lower than the first replicate of the 1.03 T and 1.07 T datasets. The difference in resonance frequencies between the midpoint of the eight replicates and the high-statistics sample was, therefore, similar to the frequency range spanned by the replicates (900 kHz). Because of this, we use the difference between our standard model and the one from the fit to the high statistics sample as a proxy for the typical variation that could be observed over the course of a measurement series.

To cross-check our estimated uncertainty, we repeated the nominal fit while removing the constraint that there is a common rise width ($\sigma$) for the eight replicates and let it vary for each. This yields values of the extracted hyperfine splitting that differ less than 2 kHz from the nominal results for both series, well within the estimated reproducibility term, with comparable statistical uncertainties.

**Signal model.** The $|c\rangle$- and $|d\rangle$-state antihydrogen atoms are obtained from the same synthesis process and are trapped in the same magnetic field profile; therefore, the lineshapes are expected to be subject to the same broadening mechanisms (motional broadening and Zeeman broadening). We also tune the injected power to match the two annihilation distributions. Barring effects associated with changing microwave powers over the frequency range of the experiment, addressed by the reproducibility systematic uncertainty, the same signal model should apply to both observed lineshapes. Because of this, and because our hyperfine splitting determination is a difference measurement, many potential systematic uncertainties associated with the signal model cancel out.

We evaluated the systematic uncertainty associated with the choice of base function by repeating the process using an alternative base model given by

$$
g(\bar{x}) \propto
\begin{cases}
0 & \bar{x} < 0, \\
\left(\dfrac{\bar{x}}{\sigma}\right)^{k} \times \exp\left\{-\left(\dfrac{\bar{x}}{\sigma}\right)^{k+1}\dfrac{\sigma}{\sigma_r}\left(\dfrac{1}{k+1}\right)\right\} & (\bar{x} \geq 0) \wedge (\bar{x} < \sigma), \\
\exp\left\{\dfrac{\sigma}{\sigma_r}\left(\dfrac{k}{k+1}\right)\right\} \times \exp\left(-\dfrac{\bar{x}}{\sigma_r}\right) & (\bar{x} \geq \sigma).
\end{cases}
$$

where $\sigma$ and $k$ are the length and sharpness of the rising edge and $\sigma_r$ describes the exponential decay. The term $\exp\left\{\frac{\sigma}{\sigma_r}\left(\frac{k}{k+1}\right)\right\}$ ensures continuity of the function at the maximum. This effect contributes 0.7 kHz to the systematic uncertainty of the hyperfine splitting at 1.03 T and 0.5 kHz at 1.07 T. We also varied the degree of the power-law rise, $k$, of the standard base function from 1 (a linear rise) to 5.4 (obtained by fits to high statistics data samples) to study the robustness of the result to the shape of the rising edge. This effect contributes 3.2 kHz and 4.2 kHz, respectively, to the systematic uncertainty at 1.03 T and 1.07 T.

We also evaluated the effect of our choice of resolution function with an alternative (one-sided cusp) model given by

$$
R(\bar{x}) \propto
\begin{cases}
\dfrac{1}{\sigma_c}\exp\left(-\dfrac{\bar{x}}{\sigma_c}\right) & \bar{x} < 0, \\
0 & (\bar{x} \geq 0).
\end{cases}
$$

Based on this analysis, we conclude that our resolution functional form choice contributes 0.6 kHz to the systematic uncertainty of the measurement at 1.03 T and 0.3 kHz at 1.07 T.

Moreover, we estimated the effect of asymmetries in the two line-shapes by varying the $\sigma_b$ parameters of the standard resolution function for the two transitions by ±2 kHz separately. This range is chosen based on fits to the simulated data over a range of microwave powers and represents a 20% difference (±2 kHz over 10 kHz) in the widths for the resolution functions between the two transitions. This effect contributes 3.5 kHz to the systematic uncertainty of the hyperfine splitting measurement at 1.03 T and 3.5 kHz at 1.07 T.

Finally, we studied the effect of varying the magnetic field profile (simultaneously for both transitions) in simulations. This resulted in a range of fitted $\sigma_b$ values between 7 kHz and 13.6 kHz but had a negligible effect on the determined hyperfine splitting.

**Binning.** Binning is a source of systematic uncertainty, quantified as $\Delta f/\sqrt{6}$, because of the discrete nature of the scan and the fixed frequency separation between the Phase 1 and Phase 3 scans of each replicate. Uncertainties from the time of flight of the antihydrogen annihilation and synchronization have also been considered, by subtracting the 30 ms after each frequency transition from the event time and repeating the analysis. The difference with respect to the nominal case is found to be negligible.

**B-drift.** Deviations from the long-time (hours) linear decay assumption of the magnetic field have been explored with Gaussian Process Regression[44] and considered in the uncertainty budget table. A limit to short-time (minutes) deviations from linearity, correlated with antihydrogen synthesis, has been obtained in auxiliary data by varying the interval between the synthesis and the spectroscopy phase of the experiment.

## Validation and cross-checks

The signal-model-fitting algorithms were validated using Monte Carlo pseudo-experiments, in which synthetic experimental lineshapes were generated from the empirical model itself. The normalized residuals of the fitted parameters confirmed that the procedure was unbiased and that the associated uncertainties were accurately estimated.

To assess the sensitivity of the measurement to residual differences between the base lineshapes for the two transitions, the analysis in Fig. 4 was repeated using the peak of the lineshapes, $f_{max} = f^{\circ} + \sigma(k+1)$, instead of the onsets $f^{\circ}$. The peak of the lineshape is expected to be more sensitive to systematic shifts because it depends on the balance of the expansion of the resonant volume (which increases the annihilation rate) and the depletion of the trapped population (which decreases the annihilation rate and depends on the microwave magnetic field strength). Both observations and simulations indicate that $f_{max}$ is more sensitive to variations in trap power between the two transitions, which in turn affects the rising edge of the signal. Nevertheless, the hyperfine splitting values obtained using $f_{max}$ are within 1.7 kHz and 0.5 kHz of our main result for the 1.03 T and 1.07 T measurements, respectively. These deviations are well within the systematic uncertainties associated with the signal model.

To verify that our treatment of the background is adequate, we examined possible deviations correlated with the axial and radial distribution of the annihilation vertices, which differ between the signal

and background, and among different sources of background. No statistically significant deviations were observed when varying the background contribution by applying tighter radial or axial selections to the annihilation events.

## Data availability

The datasets generated during and/or analysed during the current study are available from the corresponding author upon reasonable request.

## Code availability

Codes used for data analysis in the current work are available from the corresponding author upon reasonable request.

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

**Acknowledgements** This work was supported by CNPq, FAPERJ, RENAFAE (Brazil); NSERC, NRC/TRIUMF, EHPDS/EHDRS, CFI, DRAC (Canada); FNU (Nice Centre), Carlsberg Foundation (Denmark); STFC, EPSRC, the Royal Society and the Leverhulme Trust (the UK); DOE, NSF (the USA); ISF (Israel); and VR (Sweden). We are thankful for the support from the Italian PhD in Space Science and Technology (CUP E66E23000110001). We are grateful for the efforts of the CERN AD/ELENA team. We thank J. Webber (Calgary) for financial and logistical administration support. We thank the following students who contributed to this work: M. D. Agudo Barreto, A. Ferreira, A. Frayne, S. Price, E. Sweeney, H. Strojecka, U. Waheeduddin, T. Wells and H. Yaqoti.

**Author contributions** All authors are members of the ALPHA Collaboration at CERN. This experiment was based on the data collected using the ALPHA-2 antihydrogen trapping apparatus. The ALPHA-2 apparatus was designed and constructed by the ALPHA Collaboration and operated using the methods developed by the entire Collaboration. The experimental protocol was conceived by M.E.H. and T.F. and developed by T.F., M.E.H., A.J.U.J., J. Suh, A.G.S. and C.Ø.R. The microwave hardware was developed and characterized by M.E.H., T.F., A.J.U.J. and A.G.S. Electron cyclotron resonance diagnostics were developed and implemented by A.P., T.F., A.G.S., J. Suh, A.J.U.J., M.E.H. and S.G.W. Simulation of microwave interactions with the magnetically trapped atoms was developed by F.R. Trap magnet current stabilization and characterization were done by W.B., M.E.H. and P.H. The signal model and analysis procedure were developed by A.D.V., S.S. and G.B. with contributions from T.F., M.E.H., A.J.U.J., J. Suh, D.H., F.R., C.Ø.R., G.S. and L.M.G. The paper was written by T.F. and A.J.U.J. with contributions from M.E.H., G.B., F.R., J. Suh, S.S. and J.S.H. The paper was then edited and improved by all authors. In addition to those listed above, the following authors contributed to the data-taking by working shifts on the ALPHA experiment: R.A., L.O.d.A.A., C.J.B., N.M.B., A.C., I.C., C.L.C., M.C., A.C.M., D.D.Q., S.E., A.E., J.F., M.C.F., M.B.G.G., C.A.I., S.A.J., S.J., N.M., V.R.M., J.T.K.M., T.M., J.N., A.N.O., A.P., T.R.-B., R.L.S., E.S., J. Schoonwater, D.M.S., J. Singh, C.S., T.D.T., K.A.T., R.I.T., E.T.-W., M.U., D.P.v.d.W., P.W. and J.S.W.

**Competing interests** The authors declare no competing interests.

**Additional information**
**Correspondence and requests for materials** should be addressed to T. Friesen, J. S. Hangst or A. J. Uribe Jimenez.

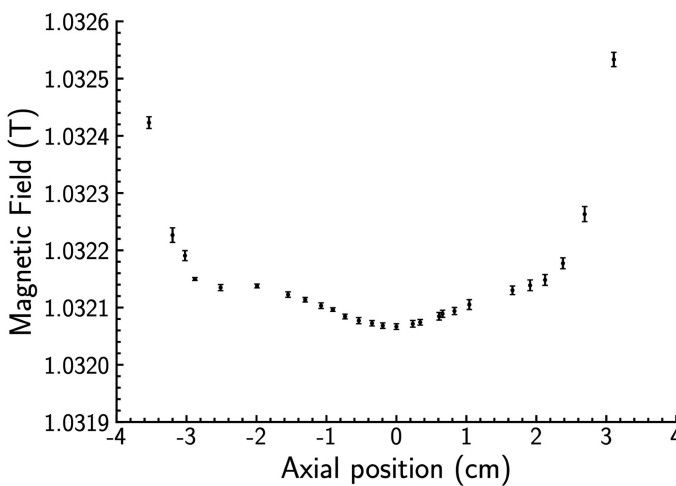

**Extended Data Fig. 1 | On-axis magnetic field profile.** The on-axis magnetic field as measured by ECR magnetometry near the axial minimum field for the 1.03 T dataset. The error bars represent the standard deviation of a Gaussian fit to the ECR signal. Uncertainty increases as the magnetic field gradient across the electron clouds increases. The positron spin-flip resonance sweeps span 240 kHz in frequency, which is equivalent to a magnetic field range of $8.6 \times 10^{-6}$ T. This ensures that the spin flips occur very close to the center of the trap.

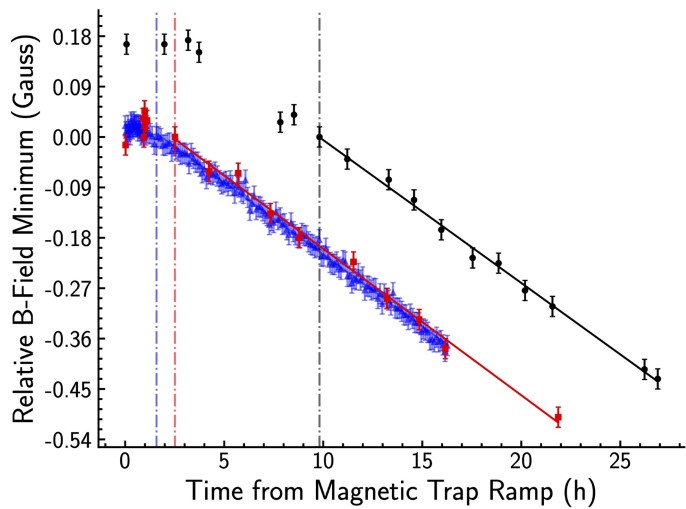

**Extended Data Fig. 2 | On-axis magnetic field minimum drift measured by ECR.** The relative on-axis magnetic field minimum measured using ECR following the ramp up of the neutral trap magnets during the 1.03 T experiment (black circles), the 1.07 T experiment (red squares), and an ECR only measurement at 1.03 T (blue triangles). The error bars represent the standard deviation of a Gaussian fit to the ECR signal. Zero on the vertical axis is taken as the first ECR measurement following the first replicate of the two experiments (marked by the dot-dashed vertical lines) and is also start of the linear fit region. The fit is chosen to start at 1.5 hours for the ECR only dataset (blue). We find a slopes of −0.025 +/− 0.001 Gauss per hour, −0.026 +/− 0.001 Gauss per hour, and −0.0255 +/− 0.0004 Gauss per hour for the 1.03 T, 1.07 T, and ECR only datasets, respectively.

## Extended Data Table 1 | Injected powers and annihilation event counts

| B-Field (T) | Injected power (mW) | | | | Total # of annihilation events | | | |
|---|---|---|---|---|---|---|---|---|
| | Phase 1 | Phase 2 | Phase 3 | Phase 4 | Phase 1 | Phase 2 | Phase 3 | Phase 4 |
| 1.03 | 50 | 140 | 40 | 130 | 5832 | 366 | 5768 | 338 |
| 1.07 | 150 | 120 | 70 | 130 | 4762 | 276 | 4581 | 255 |

Injected microwave powers and total number of annihilation events (across all 8 replicates) in each phase of the experiment. The microwave powers are referenced to the window where microwaves enter the ultra-high vacuum region of the apparatus.