## [Peer Review file · Nature]

Four ppm measurement of the antihydrogen ground state hyperfine splitting

Corresponding Author: Professor Jeffrey Hangst

Version 0:

Reviewer comments:

Referee #1

(Remarks to the Author)

The manuscript reports a 4 ppm determination of the ground-state hyperfine splitting in antihydrogen at 1 T. This represents a two-order-of-magnitude improvement over previous work and constitutes an important step forward in the precision spectroscopy of antimatter.

The work is technically impressive and scientifically solid. However, whether the article reaches the level expected for Nature is less clear. A two-order-of-magnitude improvement can indeed be considered a major advance, but the authors should further emphasize how this result changes the field and its broader implications. In addition, several aspects of the presentation and analysis require clarification. For instance:

The term CPT “absolute energy scale” is never defined. Does it refer to the energy corresponding to their frequency uncertainty? The authors should explain this concept explicitly and justify why this metric is relevant for CPT comparisons because it is a debated point because (for instance) the underlying transitions probe different operators or symmetry sectors of the Standard Model Extension.

It is unclear why Phases 1 and 3 are not randomized. Since magnetic-field drift is a key systematic, interleaving the two transitions—or at least demonstrating that the measurement order introduces no bias—would strengthen the analysis. The authors should justify their choice and ideally provide a test for order-dependence.

The choice of the $g(x)$ function (for example, the asymmetric Gaussian) is poorly explained. A brief physical justification—such as field inhomogeneity, power broadening, or depletion—should be provided. Moreover, alternative functional forms should be tested to demonstrate that the fitted transition frequency is robust.

The text mentions detailed Monte Carlo simulations used to fix parameters and validate the fitting procedure, yet no results are shown. A representative comparison between simulated and fitted spectra, or at least a distribution, is essential and currently missing.

Table 1 shows a much larger reproducibility term at 1.03 T than at 1.07 T. The origin of this discrepancy should be discussed quantitatively—for instance, in terms of differing microwave powers or coupling efficiencies.

The symbols (diamonds and squares) appear inverted relative to the caption of Figure 4; this should be verified and corrected.

The authors should provide a reference or technical description for the ultra-stable DC current–current transformer used for magnetic-field control.

Referee #2

(Remarks to the Author)

In this paper, the authors report a new measurement of the ground-state hyperfine splitting (HFS) in antihydrogen. Antihydrogen, composed of an antiproton and a positron, is the antimatter counterpart of hydrogen. Its precision

spectroscopy provides a powerful test of CPT symmetry through comparison with hydrogen.

In this experiment, a relative precision of 4 ppm was achieved for the 1S-HFS of antihydrogen. This represents a 100-fold improvement in precision compared to the result reported in 2017 and can be considered a breakthrough in antimatter science. This improvement was realized through advanced stabilization and homogenization of the trapping magnetic field, an improved antihydrogen accumulation rate, and precise control of the microwave power.

Notably, the precision achieved is sufficient to test the internal structure (finite-size effect) of the antiproton, which is a highly commendable achievement. This brings a unique value to this research, distinct from other CPT tests based on matter-antimatter comparisons. The paper is clearly structured, and it appropriately describes the excellent techniques developed by the authors and the results they have obtained.

In principle, the reviewer finds this paper to be worthy of publication in Nature. However, it is also felt that there is room for improvement in the description of several important details. Before publication, the authors are requested to address the following points.

////

1. The authors should help a broad readership understand the unique significance of this experiment, particularly in contrast to CPT tests using antihydrogen 1S-2S spectroscopy, or those comparing the antiproton magnetic moment and charge-to-mass ratio. For example, the 1S-2S transition primarily tests bound-state QED. The 1S-HFS, however, is unique in its ability to probe the internal structure of the antiproton, allowing for a comparison of matter and antimatter from the perspective of low-energy QCD.

2. The "absolute energy sensitivity" metric may be difficult for readers unfamiliar with the Standard-Model Extension to understand. Could the authors provide a plain-language explanation? A background discussion explaining "why it is evaluated on an absolute energy scale" and "what its physical meaning is" would be desirable.

3. Is there a risk of systematic uncertainties being overlooked or underestimated because the frequency scan is limited to a low-to-high direction? Are these possibilities appropriately discussed? The reviewer is concerned about effects such as an asymmetry in the atomic population's response due to the scan direction. This could bias the energy or position of the ejected antihydrogen, or introduce changes in the time-varying profile of the microwave power that atoms experience during the sequence. Would reversing the order of the frequency scan (from high-to-low) help to understand or reduce the systematic uncertainty?

4. Could a systematic uncertainty arise from assuming the microwave magnetic field is spatially uniform without discussing its actual profile? Alternatively, is the uncertainty from this simplifying assumption already accounted for within the lineshape modeling uncertainty? The authors state, for example, that f_0 is more robust than f_{\max} against changes in microwave power. This discussion should be made more quantitative.

5. The "alternative base model" mentioned in the Signal Model section lacks a specific explanation. A significant part of the systematic uncertainty is calculated by comparing the main analysis (using the empirical piecewise function $g(x)$) against this "alternative." Given its major contribution, this detail is core to the analysis. The "alternative" model must be described to justify the uncertainty estimation.

6. The justification for using the high-statistics sample to evaluate the uncertainty of the eight measurements needs to be reinforced. The reviewer suggests a simpler check: What would the result be if the data from the eight frequency scans were analyzed independently to determine f_0 ? The residuals from their weighted average could then be evaluated, after correcting for variations in measurement conditions. If this simpler analysis yields results consistent with the high-statistics reference analysis, the arguments in this paper would become exceptionally persuasive.

7. The reason why the reproducibility systematic uncertainty differs significantly between the 1.03 T and 1.07 T datasets should be explained with more specificity. If a particular factor is dominant, an explanation should be added.

8. The paper discusses prospects for reducing systematic uncertainties by employing spectroscopy at a "magic field" and using laser cooling. It would be beneficial to state the expected degree of improvement from each method. For instance, if the magic field cancels the leading order of the resonance frequency's magnetic field dependence, could the drift and reproducibility uncertainties be reduced by an order of magnitude? Making these projections (semi-)quantitative would more concretely demonstrate the project's potential.

9. Other minor comments:

9-1. Regarding the reference to the preceding paper [13], which achieved a precision of 400 ppm: would it not be more appropriate to place the citation after "400 parts-per-million" on p. 2, L. 9, rather than its current placement after "state-of-the-art" on p. 2, L. 11?

9-2. In Figure 1, some of the text, such as the labels for 'air' and 'liquid helium', is displayed in a font that is difficult to read (see an attached screenshot). While this could be an issue originating from the reviewer's environment, it might also be related to font embedding. I recommend that the authors check this.

Referee #3

(Remarks to the Author)

(A) Summary of the key results

The ALPHA collaboration improved their previous result on the antihydrogen ground state hyperfine splitting energy by a factor of 100, to ~4ppm fractional precision. This is a truly significant leap, using novel experimental techniques that has a potential to greatly impact future improvements in precision in high-field measurements.

(B) Originality and significance:

The result is original and novel, with significant improvements concerning state-of-the-art experimental techniques (magnetic field control, stabilisation and characterisation, improved accumulation rates). The present measurement has a high potential to probe the internal structure of the antiproton, the CPT invariance, and with 2S hyperfine splitting measurements and bound-state QED with a high precision.

(C) Data & methodology:

The data presented here are high quality, and the main aspects of the methodology is also clear.

The measurements are carried out in high magnetic fields which presented most of the challenges (and systematic uncertainties) in this work. However, the ALPHA collaboration demonstrated world leading control techniques over magnetic fields (few mA control over several hundreds of ampere currents) and precise studies (electron cyclotron resonance) to map these fields.

The methods are briefly represented in the online 'Methods' section and cited publications, with some information still missing for complete understanding of the present analysis, (due to spatial limitations I assume). My comments concerning the presentation of the data and calibration methods are detailed in this review, Section (F).

(D) Appropriate use of statistics and treatment of uncertainties

The approach seems solid and thorough, and key aspects were demonstrated in some earlier works. Some key information on the treatment of magnetic field drifts is missing (due to spatial limitations in the paper, I assume), but I trust the study was carried out in a thorough manner. My comments concerning the presentation of the data and calibration methods are detailed in this review, Section (F).

(E) Conclusions:

The conclusions are actually better represented somewhat better in the Abstract. This last section of the manuscript seemed a bit rushed - I suggest a rewrite here to guide the reader better concerning the impact of this work. (See in section (F) my corresponding comment.)

(F) Suggested improvements (in sequence of the paper)

Page 2

- Line 3 - the hydrogen atom is indeed pivotal for the foundations of quantum theory. Instead of writing remarkable precision', it would be prudent to add the actual fractional precision $4.2e(-15)$, and even more importantly, citing the publication, C. G. Parthey et al., Physical Review Letters (Vol. 107, 2011). Citing the historic publications are of course important too, but might be less relevant here.

- Line 36 - while the Sternheim interval offers a test of bound-state QED independent of the proton's structure uncertainties, it would be useful to point out (here, or in the conclusions) how would it contribute to higher order QED effects compared to the to 1S-2S spectroscopy antihydrogen.

Page 3

- Figure 1 labels: half of the fonts are broken. In any PDF viewer.

Page 5

- Line 16 - What does the assumption entail that the "Breit-Rabi diagram for ground-state antihydrogen is analogous in structure to that of hydrogen"? Are you assuming the exact same slope in the fit, or only the linear approximation in this field region? (Explain here or in methods)

- Line 20 - The experiments were carried out using two different magnetic fields, with ca. 4% change in the magnetic field minimum. When changing the field, a drift attributed to flux redistribution was observed. The authors were waiting 1.5 h before starting the measurement cycle to ensure a linear drift - (74 kHz/h)? I assume this is a fit that comes from multiple

ECR measurements (confirming the reproducibility in the field cycling methods), but more details are needed in Methods (See also my comment later)

- Line 40 - After c->b spectroscopy (Phase 1), the remaining c-state antihydrogen is removed (Phase 2) in order to “decrease the background” before d->a spectroscopy. After the d->a measurement (Phase 3) the remaining d-states are also removed (Phase 4). It is not mentioned in the main text whether the annihilations measured during these removal phases (Phases 2, 4) are used for the analysis (which I believe is the case, to measure the state populations), so it is unclear why Phase 4 is needed. Please add more clarity to the description of these experimental sequences.

Page 6

- Figure 3 - In the Y axis, the label ‘Events’ are not explained, (I assume these are normalised events to the populations in c and d states? Something more complicated?).

Page 7

- (Or in Methods). A discussion about the possible differences in the empirical fits you be interesting to understand for the two frequencies. For instance, it is not clear to me how this larger width of the empirical fit for d->a sweep and c->b sweep is characterised (See my comments to Methods). In essence, this part of the Main text is hard to read without the Methods, and the methods on this part are unclear as well. I suggest extending the information on this function fit, and have a better reference to in Main.

- Figure 4 (and explaining text): the drifting magnetic field has a huge effect on the measured frequencies (1 GHz on this whole scale) while the determined uncertainty for a1S/h is on the kHz level. In my understanding (Methods), this drift was characterised in the axial direction using ECR methods, but no plot or fit is accompanying this (See my comments to Methods). I suggest to extend Methods with this information, and refer to in Main.

Page 8

- Line 9 - It is left somewhat vague how the measurement will have an impact on the mentioned nuclear structure effects: the expression “exposes the regime” to nuclear effects should be explained here, (in terms of the Zemach radius? It is indeed on the ~50 kHz level - or some other models considered?).

As of the Sternheim interval, a more detailed description on the impact of this measurement in comparison with (the much more precise) hydrogen Sternheim interval measurements would be very welcome, also in light of 1S-2S spectroscopy of both atoms.

Finally, a small comment on the author list/contributions: I was sad to hear that Joel Fajans passed away last year. Of course there is no doubt that his contributions were significant to the present work, but a footnote on this would be appropriate (and a mention in the author’s contributions.)

Methods:

Page 9

The magnetic field characterisation is the most important part of the systematic error studies.

The shallow centrally located minimum in the axial field (instead of a maximally flattened one in the previous works) was mapped by ECR technique, and the drifts of this field during the magnet sequencing was characterised.

I assume this would entail a detailed study on the observed flux relaxation and the linear drift phase in which the experiment was carried out - which is extremely important for the analysis. How reproducible is the slope? It is mentioned that it ‘depends a lot’ on the history of the magnet, which is indeed expected. Hence, the same sequencing in the magnet was used for all of the measurements.

Please show in a figure a set of these drift plots, during multiple typical experimental sequences, and the fit on the linear drift in the field. This would be pivotal here to show the reproducibility of the B field sequence, as the gradient is used in the analysis.

Page 12

A bit unclear to me how the width of the two (c->b and d->a) base functions were determined. If I understood correctly, $k=2$ and $\sigma=10$ kHz is the same for both transitions, I assume (from your MC), but (line 16-17) says “The parameters controlling the lengths of the rising and falling edges of the empirical base function were allowed to differ”. How were the residual spin-flip rates determined for these parameters? I’m assuming here that you use the ‘cleanup’ cycles of the remaining c and d states to do this, but I don’t see it mentioned here.

(G) References

The manuscript is generally well referenced, however I find that if historic references are used for the hydrogen atom, (questionable if all is necessary) one is not allowed to miss the most important modern precision reference, the hydrogen 1S-2S measurement (C. G. Parthey et al., Physical Review Letters (Vol. 107, 2011) At least this should be added in the appropriate place (see point F). A general revision of the relevance of some of these references would be advised.

(H) Clarity and context

The abstract and introduction are appropriate and lucid, and the conclusions appropriately involve the context of other research groups. The conclusions miss some more factual details (as explained in Section F), and the style of the writing could be greatly improved in this last part.

Version 1:

Reviewer comments:

Referee #1

(Remarks to the Author)
recommends publication

Referee #2

(Remarks to the Author)

The authors have addressed the reviewers' comments appropriately and revised the manuscript accordingly. The lack of detail present in the previous version has been resolved, and the reviewer now considers the content suitable for publication in Nature.

Precision physics with antimatter is a powerful and irreplaceable tool for understanding the origins of our universe. By reporting a significant breakthrough in this field with such rigor, this manuscript is well-suited for publication in Nature and will undoubtedly be of great interest to a broad audience beyond the immediate community.

Referee #3

(Remarks to the Author)

The referee appreciates the authors' thorough response to the raised issues and the implemented corrections.

The clarification of the measurement sequence and the description of the PSR/ECR method improve the transparency of both the measurement procedure and the treatment of systematic uncertainties. Extended Data Figure 2 and the connecting text now more clearly explains the treatment magnetic field drift. (Although the MHz/GHz was clearly a typo on my part, the drift itself is substantial and its treatment is important.) The revised manuscript reads more clearly overall, with the expanded Methods section notably improving the presentation of the chosen signal model and the Monte Carlo simulations as well.

Assessing systematic uncertainties in high-field measurements is inherently challenging, and admittedly, similarly challenging to evaluate all details of it in this short paper. While the manuscript conveys a careful systematic treatment and control over the experimental parameters, future 'zero-field' (beam) measurements are encouraged to offer an independent precision benchmark.

I would reiterate that this result represents a significant, two-order-of-magnitude advance in (high-B-field) measurements of the ground state hyperfine splitting of antihydrogen. This result may, for the first time, provide access to information on the magnetic structure of the antiproton. The results presented here are unparalleled so far in antihydrogen spectroscopy, and highly relevant for precision physics. Hence I endorse its publication in Nature.

Referee #1 (Remarks to the Author):

The manuscript reports a 4 ppm determination of the ground-state hyperfine splitting in antihydrogen at 1 T. This represents a two-order-of-magnitude improvement over previous work and constitutes an important step forward in the precision spectroscopy of antimatter.

The work is technically impressive and scientifically solid. However, whether the article reaches the level expected for Nature is less clear. A two-order-of-magnitude improvement can indeed be considered a major advance, but the authors should further emphasize how this result changes the field and its broader implications.

In addition, several aspects of the presentation and analysis require clarification. For instance:

The term CPT “absolute energy scale” is never defined. Does it refer to the energy corresponding to their frequency uncertainty? The authors should explain this concept explicitly and justify why this metric is relevant for CPT comparisons because it is a debated point because (for instance) the underlying transitions probe different operators or symmetry sectors of the Standard Model Extension.

Author Response: We thank the referee for this excellent feedback and important comment. The original discussion regarding the absolute energy scale was motivated by CPT-violating theories, based on an effective field theory framework such as the Standard Model Extension. However, a detailed interpretation of our results within this context is beyond the scope of the present article. We have therefore removed this discussion and instead focus on other aspects, such as the internal structure of the antiproton. Concerning the article’s acceptability in Nature, we address some comments to the Editor in the Cover Letter.

Revisions: We have removed the sentence referring to the absolute energy scale from the abstract and phrased the sentence in the introductory paragraph to remove reference to absolute energy comparisons.

It is unclear why Phases 1 and 3 are not randomized. Since magnetic-field drift is a key systematic, interleaving the two transitions—or at least demonstrating that the measurement order introduces no bias—would strengthen the analysis. The authors should justify their choice and ideally provide a test for order-dependence.

Author Response: Phase 1 ($|c\rangle \rightarrow |b\rangle$ spectroscopy) and Phase 3 ($|d\rangle \rightarrow |a\rangle$ spectroscopy) are performed in this order because the higher-frequency microwaves in Phase 3 could induce

$|c\rangle \rightarrow |b\rangle$ transitions in antihydrogen atoms with sufficient kinetic energy to reach regions of high magnetic field. This would eject the majority of the $|c\rangle$ -state atoms from the trap during Phase 3 and create a very high annihilation background when trying to determine the onset of annihilations from $|d\rangle \rightarrow |a\rangle$ transitions. In the adopted protocol, the possibility of contamination from $|c\rangle \rightarrow |b\rangle$ transitions during $|d\rangle \rightarrow |a\rangle$ spectroscopy is mitigated by Phase 2, during which residual $|c\rangle$ -state atoms are cleared. Any remaining $|c\rangle$ -state atom annihilating during Phase 3 is accounted for by the background component in the fit.

As the referee correctly points out the magnetic-field drift is a key systematic we account for this by taking eight replicates for each magnetic field setting (1.03 T or 1.07 T), and rather than using each replicate to extract the hyperfine splitting we take the difference of the two lines (Figure 4) to extract the hyperfine splitting. We account for possible effects due to magnetic-field drifts deviating from linearity on the short time-scale corresponding to the separation between Phase 1 and Phase 3 as a systematic uncertainty. This is done by performing a set of repetitions consisting of two interleaved series that differ by the wait time (32 s and 352 s, respectively) between antihydrogen synthesis and Phase 1 spectroscopy. We then compare the hyperfine splitting measurements obtained using different combinations of Phases 1 and 3 from the two different series (with 32 s and 352 s waits), so that short-time magnetic field drifts would affect each combination differently. From this comparison we derive a systematic uncertainty of 1.1 kHz that contributes to the B-drift systematic uncertainties reported in Table 1 for both series.

Revisions: We have added the following sentence to the end of the first paragraph of “The spectroscopy experiment” to clarify why we cannot invert the order in which we drive the transitions.

“...splitting frequency a_{1S}/h . Note that the order in which transitions are driven cannot be reversed (ie. $|d\rangle \rightarrow |a\rangle$ followed by $|c\rangle \rightarrow |b\rangle$) because $|c\rangle$ -state atoms at high magnetic fields will come into resonance when driving $|d\rangle \rightarrow |a\rangle$ transitions near the magnetic minimum. This will remove a significant number of $|c\rangle$ -state atoms from the trap and create a high annihilation background that will mask the onset of $|d\rangle$ -state atom annihilations. “

The choice of the $g(x)$ function (for example, the asymmetric Gaussian) is poorly explained. A brief physical justification—such as field inhomogeneity, power broadening, or depletion—should be provided. Moreover, alternative functional forms should be tested to demonstrate that the fitted transition frequency is robust.

Author Response: Because the chosen model is ultimately empirical, due to lack of complete knowledge regarding full magnetic trap and microwave field conditions, we cannot include a true physical justification of the $g(x)$ function. The guiding principles, however, are discussed in the “Data analysis” section of the Methods. As described in the Analysis and Results section of the main text, we are performing a difference measurement and systematic effects associated with our choice of model largely cancel out, and we quantify the residual systematics associated with our choices.

The rising part of the distribution, where depletion has not yet set in, is dominated by the increase in the resonant volume (the region where the magnetic field value corresponds to a transition frequency equal to the applied microwave frequency) as the frequency is increased. This is modelled as a polynomial component that could be interpreted as the first term of a Taylor series approximation of the resonant volume as a function of frequency.

The asymmetric-Gaussian core simply provides sufficient capacity to satisfactorily adapt to the transition between the rise and the tail of the experimental distributions, which is dominated by the depletion of the anti-hydrogen population. We find that the rise and tail of the annihilation distributions have different scales, hence the choice of an asymmetric Gaussian, whose left- and right-side widths (σ and σ_r , respectively) are both free parameters of the model.

We evaluated the systematic uncertainty associated with the choice of the functional form by repeating the fit procedure using the following alternative base model:

$$g(\bar{x}) \propto \begin{cases} 0 & \bar{x} < 0, \\ \left(\frac{\bar{x}}{\sigma}\right)^k \times \exp\left\{-\left(\frac{\bar{x}}{\sigma}\right)^{k+1} \frac{\sigma}{\sigma_r} \left(\frac{1}{k+1}\right)\right\} & (\bar{x} \geq 0) \wedge (\bar{x} < \sigma), \\ \exp\left\{\frac{\sigma}{\sigma_r} \left(\frac{k}{k+1}\right)\right\} \times \exp\left(-\frac{\bar{x}}{\sigma_r}\right) & (\bar{x} \geq \sigma). \end{cases}$$

The use of this alternative model for g , characterized by a different parametrization of the core and tail of the distribution, results in extracted values for the hyperfine splitting that differ from the nominal ones by 0.7 kHz and 0.5 kHz (for the 1.03 T series and 1.07 T series, respectively). This forms part of the systematic uncertainty associated with our choice of signal model, which overall contributes 4.8 kHz and 5.5 kHz (for 1.07 T and 1.03 T series, respectively) to the uncertainty of our hyperfine determination. Other factors entering this uncertainty are described below.

We also studied the robustness of the hyperfine splitting result to the shape of the rising edge by repeating the fit with the degree k of the polynomial varying between 1 and 5.4 (the range of values observed in high-statistics data samples, whereas $k = 2$ is the value used in the nominal fit). From these we derive a systematic uncertainties of 4.2 kHz (1.07 T series) and

3.2 kHz (1.03 T series). These two effects are described (albeit without specifying the alternative $g(x)$ function) in Systematic Uncertainties - Signal Model of the Methods.

We also altered the shape of the resolution function (which is motivated by potential physical broadening effects) to further study the effect of our chosen empirical model. We altered the shape of the resolution function from a one-sided Gaussian as defined in the Data Analysis section of Methods to a one-sided cusp given by:

$$R(\bar{x}) \propto \begin{cases} \frac{1}{\sigma_c} \exp\left(-\frac{\bar{x}}{\sigma_c}\right) & \bar{x} < 0, \\ 0 & (\bar{x} \geq 0). \end{cases}$$

Based on this analysis we conclude that our resolution functional form choice contributes a 0.6 kHz (0.3 kHz) to the systematic uncertainty of the measurement at 1.03 T (1.07 T). We also studied the effect of our resolution function by varying the width of the original Gaussian resolution function between 7 and 13.6 kHz but this had a negligible effect on the determined hyperfine splitting.

Finally, the main systematic uncertainty associated with our empirical function is due to potential asymmetries in the two lineshapes that can be quantified by assuming a 20% difference (± 2 kHz over 10 kHz) in the widths for the resolution functions between the two transitions. This effect contributes 3.5 kHz uncertainty to both series. These effects are described (again without specifying the alternative functional form of $R(\bar{x})$ explicitly) in the Systematic Uncertainties - Signal Model of the Methods.

Revisions:

We have revised the Systematic Uncertainties – Signal Model section of the Methods section to provide more detail on our quantification of the systematic uncertainties related to our choice of the empirical function including giving the functional forms of the alternative base and resolution functions studied.

The text mentions detailed Monte Carlo simulations used to fix parameters and validate the fitting procedure, yet no results are shown. A representative comparison between simulated and fitted spectra, or at least a distribution, is essential and currently missing.

Author Response: We use two types of “simulations” in our analysis.

The first type is described in the Simulations section of Methods, which is used to fix two parameters of our model. It attempts to simulate the motion of antihydrogen atoms in the magnetic trap and the interaction with microwaves. This simulation is computationally expensive and cannot fully model the observed lineshapes at the required precision due to lack of knowledge of the microwave field and the static magnetic field off-axis. However, they do allow us to study the sensitivity of the lineshapes to different microwave powers and

magnetic field profiles. These simulations combined with high statistics data samples are used to fix the degree of the power-law rise, $k = 2$, and the resolution function broadening parameter $\sigma_b = 10$ kHz. The uncertainty with these choices is then quantified in the Systematic uncertainties section of Methods.

The other type is referred to in the text as a “Monte Carlo pseudo-experiment” and is used to validate the fitting procedure. These Monte Carlo pseudo-experiments generate a large number of annihilation distributions from the fit model itself and do not simulate the physics of the experiment. We use this to study the statistical properties of the fit, such as the stability of the fit configurations, the reliability of the uncertainty estimates, and whether the fitting procedure itself introduces a bias.

It’s important to clarify that these are two separate analyses used for different purposes. We interpret this comment as being primarily about the first type of simulations. We presume the referee is looking for a comparison of the empirical function obtained when fitted to simulated data versus that obtained with fits to experimental data. It should be re-emphasized that we rely very little on the simulations and their shortcomings are discussed in the paper and associated systematic uncertainties are accounted for. Because of this, we do not feel that including a figure in the article that compares simulated data and experimental data provides essential background or information.

Revisions: We have added a clarification to the introduction of the Monte Carlo pseudo-experiments in the Validation and cross-checks section of Methods to clarify that the synthetic lineshapes are generated from the empirical model itself.

Table 1 shows a much larger reproducibility term at 1.03 T than at 1.07 T. The origin of this discrepancy should be discussed quantitatively—for instance, in terms of differing microwave powers or coupling efficiencies.

Author Response: The reproducibility contribution to systematic uncertainties addresses the possibility that the annihilation distributions vary across the eight replicates, whereas the nominal fit assumes they are the same.

A possible cause for different annihilation distributions across the eight repetitions could be a variation in the microwave power delivered to the anti-hydrogen atoms as a function of the microwave frequency (because the solenoid magnetic field is decaying, the injected microwave frequencies vary within a series, by roughly 900 kHz over a 12-hour period). This is currently discussed in the Systematic uncertainties section of Methods.

While a larger variation of power over the frequency range in each series is a plausible explanation for the observed difference in the estimated systematic uncertainty, the non-trivial dependence on frequency of the power delivered to the trapping region - due to the trap

geometry and standing wave patterns – prevents us from positively identifying this as the sole effect and from quantifying its contribution by auxiliary measurements.

We also performed a cross-check of the estimated systematic uncertainty, by repeating the nominal fit while relaxing the constraint that the shapes of the distributions should be the same across all repetitions. Doing so yields values of the extracted hyperfine splitting that are approximately 2 kHz lower than the nominal results, for both series, with comparable statistical uncertainties. This difference is within the estimated reproducibility term.

Revisions: We have added two sentences to the final paragraph of “Analysis and results” to clarify how differing microwave powers enter our uncertainty analysis through the reproducibility systematic and that this is the potential reason for the larger reproducibility uncertainty in the 1.03 T experiment. We have also added a paragraph above the above cross-check to the Reproducibility systematic uncertainty discussion in Methods.

The symbols (diamonds and squares) appear inverted relative to the caption of Figure 4; this should be verified and corrected.

Author Response + revisions: We thank the referee for catching this error. The figure caption text has been fixed to properly match the figure.

The authors should provide a reference or technical description for the ultra-stable DC current–current transformer used for magnetic-field control.

Author response + revisions: We have added the following technical description of the DCCT stabilisation system:

“During our experiments, the mirror coils are provided with currents ranging from a few amperes to almost 500 A. The octupole is operated with a current of 900 A. These currents are individually monitored using high-precision, ultra-stable direct current - current transformers based on closed-loop fluxgate magnetometer sensors (ITZ 2000-SB FLEX ULTRASTAB from LEM International SA). The DCCT provides a ± 10 V output with a 500 kHz small-signal bandwidth. This signal was digitized by 24-bit National Instruments NI-9239 cRIO (Compact Real-time Input Output) analog-to-digital converter modules at 50 kS/s and averaged by the cRIO FPGA (Field Programmable Gate Array) firmware to 10 kS/s. The resulting averaged signal was used for proportional–integral–derivative (PID)-based closed-loop control of the magnet power supplies, stabilising output currents to within several mA.”

Referee #2 (Remarks to the Author):

In this paper, the authors report a new measurement of the ground-state hyperfine splitting (HFS) in antihydrogen. Antihydrogen, composed of an antiproton and a positron, is the antimatter counterpart of hydrogen. Its precision spectroscopy provides

a powerful test of CPT symmetry through comparison with hydrogen.

In this experiment, a relative precision of 4 ppm was achieved for the 1S-HFS of antihydrogen. This represents a 100-fold improvement in precision compared to the result reported in 2017 and can be considered a breakthrough in antimatter science. This improvement was realized through advanced stabilization and homogenization of the trapping magnetic field, an improved antihydrogen accumulation rate, and precise control of the microwave power.

Notably, the precision achieved is sufficient to test the internal structure (finite-size effect) of the antiproton, which is a highly commendable achievement. This brings a unique value to this research, distinct from other CPT tests based on matter-antimatter comparisons. The paper is clearly structured, and it appropriately describes the excellent techniques developed by the authors and the results they have obtained.

In principle, the reviewer finds this paper to be worthy of publication in Nature. However, it is also felt that there is room for improvement in the description of several important details. Before publication, the authors are requested to address the following points.

////

1. The authors should help a broad readership understand the unique significance of this experiment, particularly in contrast to CPT tests using antihydrogen 1S-2S spectroscopy, or those comparing the antiproton magnetic moment and charge-to-mass ratio. For example, the 1S-2S transition primarily tests bound-state QED. The 1S-HFS, however, is unique in its ability to probe the internal structure of the antiproton, allowing for a comparison of matter and antimatter from the perspective of low-energy QCD.

Author response + revisions: We thank the referee for this excellent suggestion. We have rephrased some of the introductory paragraph to focus more on the nuclear structure sensitivity of the 1S-HFS as a complementary and contrasting measurement to 1S – 2S spectroscopy.

2. The "absolute energy sensitivity" metric may be difficult for readers unfamiliar with the Standard-Model Extension to understand. Could the authors provide a plain-

language explanation? A background discussion explaining "why it is evaluated on an absolute energy scale" and "what its physical meaning is" would be desirable.

Author Response: We thank the referee for this excellent feedback and important comment. The original discussion regarding the absolute energy scale was motivated by CPT-violating theories, based on an effective field theory framework such as the Standard Model Extension. However, a detailed interpretation of our results within this context is beyond the scope of the present article. We have therefore removed this discussion and instead focus on other aspects, such as the internal structure of the antiproton.

Revisions: We have removed the sentence referring to the absolute energy scale from the abstract and phrased the sentence in the introductory paragraph to remove reference to absolute energy comparisons.

3. Is there a risk of systematic uncertainties being overlooked or underestimated because the frequency scan is limited to a low-to-high direction? Are these possibilities appropriately discussed? The reviewer is concerned about effects such as an asymmetry in the atomic population's response due to the scan direction. This could bias the energy or position of the ejected antihydrogen, or introduce changes in the time-varying profile of the microwave power that atoms experience during the sequence. Would reversing the order of the frequency scan (from high-to-low) help to understand or reduce the systematic uncertainty?

Author Response: Our measurement requires that we scan from low to high frequencies. The basis of our measurement is identifying the positron spin flip resonance frequencies at the minimum of the magnetic field and extracting the hyperfine splitting by finding the difference at that well defined magnetic field. This is done by scanning from low to high frequencies because when the microwave drive comes into resonance with atoms at the minimum magnetic field any atom that transits the magnetic minimum can be spin flipped and annihilate, providing a clear signal. If we were to start at high frequencies, the majority of the atoms would undergo a spin flip in the first few frequency steps and by the time we reach the frequency resonant only at the magnetic minimum (the frequency point we are attempting to identify) only a vanishingly tiny number of very cold anti-atoms will remain preventing us from easily distinguishing this disappearance of signal from background. So while scanning from high to low can potentially provide some interesting information about the energy distribution of the anti-atoms it cannot be used to extract the hyperfine splitting using the method we lay out in the paper.

Revisions: We have added text to the first paragraph in "The spectroscopy experiment" section of the main text that clarifies this idea and the need to sweep from low-to-high frequencies and the need to address the $|c\rangle \rightarrow |b\rangle$ transition first.

4. Could a systematic uncertainty arise from assuming the microwave magnetic field is spatially uniform without discussing its actual profile? Alternatively, is the uncertainty from this simplifying assumption already accounted for within the lineshape modeling uncertainty? The authors state, for example, that f_0 is more robust than f_{max} against changes in microwave power. This discussion should be made more quantitative.

Author Response:

The incomplete knowledge of the spatial dependence of the microwave field is acknowledged in the choice of an empirical signal model and in the systematic uncertainties that fall into the “signal model” and “lineshape reproducibility” categories. To evaluate these systematic uncertainties, the functional forms of the model and the values of fixed parameters in the fit are varied.

It is also important to note that we do not assume the microwave magnetic field is spatially uniform. A spatially non-uniform but constant in frequency microwave magnetic field structure will result in the correct extraction of the hyperfine splitting from our difference measurement. However, the microwave magnetic field structure is not constant in frequency. We accounted for this first by tuning the injected power so that rate at which anti-atoms are ejected was closely matched between the two transitions. We then accounted for any remaining asymmetry between the two transitions in our modelling uncertainty by allowing some model parameters to vary separately (described in Methods – Systematic Uncertainties – Signal Model). Additionally, the effect of changes in microwave magnetic field over the course of each experiment was estimated by repeating the procedure using fit parameters fixed by a high statistics sample at a lower frequency range (described in Methods - Systematic Uncertainties – Reproducibility).

Furthermore, the repetition of the measurement at two different magnetic fields settings, as well as the magnetic field drift over time, mitigate the risk of pathological spatial distributions of the microwave magnetic fields associated to a particular microwave frequency, and offer a means of cross-checking the behaviour of the measurement across a wide range of frequencies.

Regarding f_0 versus f_{max} : The onset frequency f_0 was chosen as an indicator of the frequency associated with the lowest value of the static magnetic field in the trap and should be less sensitive to the shape of the magnetic trap and the depletion of the trapped population. The position of the peak frequency f_{max} , on the other hand, depends on the balance between the expansion of the resonant volume (which increases the annihilation rate) and the depletion of the trapped population (which decreases the annihilation rate, and depends on the microwave power delivered to the antihydrogen atoms). Nevertheless, the fact that we obtain the same hyperfine splitting results well within uncertainties (within 1.7 kHz and 0.5 kHz) is a testament

to the fact that by taking the difference of the onset frequencies we are not introducing a significant bias.

Revisions: To clarify that we handle the uncertainty in the microwave magnetic field structure within the Reproducibility systematic we have mentioned that effect explicitly in the main text in the final paragraph of “Analysis and results”. We have also rephrased the first paragraph of the Reproducibility part of the systematic uncertainty discussion in Methods. We have also added a sentence to in “Validations and cross-checks” in Methods to elaborate on f_o versus f_{max} .

5. The "alternative base model" mentioned in the Signal Model section lacks a specific explanation. A significant part of the systematic uncertainty is calculated by comparing the main analysis (using the empirical piecewise function $g(x)$) against this "alternative." Given its major contribution, this detail is core to the analysis. The "alternative" model must be described to justify the uncertainty estimation.

Author Response + Revisions: We thank the referee for this suggestion and agree that our treatment of the alternative model lacked detail. We have revised the Systematic Uncertainties – Signal Model section of the Methods section to provide more detail on our quantification of the systematic uncertainties related to our choice of the empirical function including giving the functional forms of the alternative base and resolution functions studied.

The analysis evaluated the systematic uncertainty associated with the choice of the base functional form by repeating the fit procedure using the following alternative base model:

$$g(\bar{x}) \propto \begin{cases} 0 & \bar{x} < 0, \\ \left(\frac{\bar{x}}{\sigma}\right)^k \times \exp\left\{-\left(\frac{\bar{x}}{\sigma}\right)^{k+1} \frac{\sigma}{\sigma_r} \left(\frac{1}{k+1}\right)\right\} & (\bar{x} \geq 0) \wedge (\bar{x} < \sigma), \\ \exp\left\{\frac{\sigma}{\sigma_r} \left(\frac{k}{k+1}\right)\right\} \times \exp\left(-\frac{\bar{x}}{\sigma_r}\right) & (\bar{x} \geq \sigma) \end{cases}$$

The use of this alternative model for g , characterized by a different parametrization of the core and tail of the distribution, results in extracted values for the hyperfine splitting that differ from the nominal ones by 0.5 kHz and 0.7 kHz (for the 1.07 T series and 1.03 T series, respectively).

6. The justification for using the high-statistics sample to evaluate the uncertainty of the eight measurements needs to be reinforced. The reviewer suggests a simpler check: What would the result be if the data from the eight frequency scans were analyzed independently to determine f_0 ? The residuals from their weighted average could then

be evaluated, after correcting for variations in measurement conditions. If this simpler analysis yields results consistent with the high-statistics reference analysis, the arguments in this paper would become exceptionally persuasive.

Author Response: The referee suggestion is indeed very pertinent.

To improve the stability of the fit, as studied using Monte-Carlo pseudo-experiments, the parameters of the fit describing the rising and falling edges of annihilation distributions were in common for all the repetitions of a given transition. As reported in the article, a systematic uncertainty is then estimated from the change in the extracted hyperfine values when replacing the model fitted on the eight measurements with a model derived from a high-statistics sample collected at a different magnetic field.

Since the difference in magnetic field between the centre of the eight measurements and the high statistics sample is similar to the magnetic field range spanned by the eight measurements, the difference between these two models is used as a proxy for the typical variation that could be observed over the course of a measurement series.

As a cross-check, the assumption that the shape remains the same was relaxed and the eight repetitions were allowed to have different rise widths (controlled by the σ parameter). While the tail parameters are still shared across the eight repetitions, this configuration is very similar to the one for the cross-check suggested by the reviewer, and results in much smaller correlation between the f_0 results from different repetitions. With this configuration, the hyperfine splitting values differ less than 2 kHz from the nominal results for both series, with a similar statistical uncertainty. These differences are well within the systematic uncertainties evaluated by the method described in the article. The residuals from the average in this configuration and the ones obtained in the nominal configuration have a correlation coefficient of 0.45 and 0.72 for the 1.03 T and 1.07 T series, respectively.

In a variation of the cross-check described above, we allow both the rise and tail widths (controlled by the σ and σ_r parameters) to vary across repetitions. In this case, to ensure the convergence of the fits we constrain the difference between the onset frequencies in Phase 1 and Phase 3 to be the same for all repetitions. The fitted values for the onset frequency difference obtained in this configuration and the average of the eight onset frequency differences obtained in the nominal configuration differ by 1.2 kHz and 0.6 kHz for the 1.07 T and 1.03 T series, respectively, i.e., they are well within the “reproducibility” systematic uncertainties reported in Table 1.

Revisions: We have added some text to justify our use of the high statistics samples for estimating the reproducibility uncertainty to Methods -Systematic Uncertainties – Reproducibility. We have also included description of the first cross-check described above that is in line with what the referee suggests.

7. The reason why the reproducibility systematic uncertainty differs significantly

between the 1.03 T and 1.07 T datasets should be explained with more specificity. If a particular factor is dominant, an explanation should be added.

Author Response: The reproducibility contribution to systematic uncertainties addresses the possibility that the annihilation distributions vary across the eight replicates, whereas the nominal fit assumes they are the same.

A possible cause for different annihilation distributions across the eight repetitions could be a variation in the microwave power delivered to the anti-hydrogen atoms as a function of the microwave frequency (because the solenoid magnetic field is decaying, the injected microwave frequencies vary within a series, by roughly 900 kHz over a 12-hour period). This is currently discussed in the Systematic uncertainties section of Methods.

While a larger variation of power over the frequency range in each series is a plausible explanation for the observed difference in the estimated systematic uncertainty, the non-trivial dependence on frequency of the power delivered to the trapping region - due to the trap geometry and standing wave patterns - prevents us from positively identifying this as the sole effect and from quantifying its contribution by auxiliary measurements.

We also performed a cross-check of the estimated systematic uncertainty, by repeating the nominal fit while relaxing the constraint that the shapes of the distributions should be the same across all repetitions. Doing so yields values of the extracted hyperfine splitting that are approximately 2 kHz lower than the nominal results, for both series, with comparable statistical uncertainties. This difference is within the estimated reproducibility term.

Revisions: We have added two sentences to the final paragraph of “Analysis and results” to clarify how differing microwave powers enter our uncertainty analysis through the reproducibility systematic and that this is the potential reason for the larger reproducibility uncertainty in the 1.03 T experiment. As discussed in response to the previous question, we have also added a paragraph above the above cross-check to the Reproducibility systematic uncertainty discussion in Methods.

8. The paper discusses prospects for reducing systematic uncertainties by employing spectroscopy at a "magic field" and using laser cooling. It would be beneficial to state the expected degree of improvement from each method. For instance, if the magic field cancels the leading order of the resonance frequency's magnetic field dependence, could the drift and reproducibility uncertainties be reduced by an order of magnitude? Making these projections (semi-)quantitative would more concretely demonstrate the project's potential.

Author Response + Response: We thank the referee for their suggestion. We expect that the NMR transition measured at the magic field of 0.65 T, combined with laser and adiabatic cooling, will enable another factor of 2 improvement in the determination of the hyperfine splitting. This is largely due to suppression of systematic effects connected to variations (spatially and temporarily) in the magnetic field as well as reduced motional effects. We have expanded on our statement in the conclusions to be (semi-)quantitative about this expectation.

9. Other minor comments:

9-1. Regarding the reference to the preceding paper [13], which achieved a precision of 400 ppm: would it not be more appropriate to place the citation after "400 parts-per-million" on p. 2, L. 9, rather than its current placement after "state-of-the-art" on p. 2, L. 11?

Author response + Revisions: We thank the referee for catching this detail and have moved the reference to the more appropriate location.

9-2. In Figure 1, some of the text, such as the labels for 'air' and 'liquid helium', is displayed in a font that is difficult to read (see an attached screenshot). While this could be an issue originating from the reviewer's environment, it might also be related to font embedding. I recommend that the authors check this.

Author response + Revisions: Our apologies for the formatting/font issue that seems to have appeared at some stage of the process. This will be corrected in the final version.

Referee #3 (Remarks to the Author):

(A) Summary of the key results

The ALPHA collaboration improved their previous result on the antihydrogen ground state hyperfine splitting energy by a factor of 100, to ~4ppm fractional precision. This is a truly significant leap, using novel experimental techniques that has a potential to greatly impact future improvements in precision in high-field measurements.

(B) Originality and significance:

The result is original and novel, with significant improvements concerning state-of-the-art experimental techniques (magnetic field control, stabilisation and characterisation,

improved accumulation rates). The present measurement has a high potential to probe the internal structure of the antiproton, the CPT invariance, and with 2S hyperfine splitting measurements and bound-state QED with a high precision.

(C) Data & methodology:

The data presented here are high quality, and the main aspects of the methodology is also clear.

The measurements are carried out in high magnetic fields which presented most of the challenges (and systematic uncertainties) in this work. However, the ALPHA collaboration demonstrated world leading control techniques over magnetic fields (few mA control over several hundreds of ampere currents) and precise studies (electron cyclotron resonance) to map these fields.

The methods are briefly represented in the online 'Methods' section and cited publications, with some information still missing for complete understanding of the present analysis, (due to spatial limitations I assume). My comments concerning the presentation of the data and calibration methods are detailed in this review, Section (F).

(D) Appropriate use of statistics and treatment of uncertainties

The approach seems solid and thorough, and key aspects were demonstrated in some earlier works. Some key information on the treatment of magnetic field drifts is missing (due to spatial limitations in the paper, I assume), but I trust the study was carried out in a thorough manner. My comments concerning the presentation of the data and calibration methods are detailed in this review, Section (F).

(E) Conclusions:

The conclusions are actually better represented somewhat better in the Abstract. This last section of the manuscript seemed a bit rushed - I suggest a rewrite here to guide the reader better concerning the impact of this work. (See in section (F) my corresponding comment.)

(F) Suggested improvements (in sequence of the paper)

Page 2

- Line 3 - the hydrogen atom is indeed pivotal for the foundations of quantum theory. Instead of writing remarkable precision', it would be prudent to add the actual

fractional precision $4.2e(-15)$, and even more importantly, citing the publication, C. G. Parthey et al., Physical Review Letters (Vol. 107, 2011). Citing the historic publications are of course important too, but might be less relevant here.

Author response + revisions: We thank the referee for the suggestion. It was an oversight on our part not to have a citation to C. G. Parthey et al., Physical Review Letters (Vol. 107, 2011) in our submission. We have re-arranged some of the references to more appropriate locations and added a citation to the aforementioned paper. Regarding adding the actual fractional precision to the second sentence abstract, we feel that this change does not strengthen the abstract or paper. That sentence is meant to reference not only to the 1S – 2S transition frequency measurement but also to hyperfine and other spectroscopy results.

- Line 36 - while the Sternheim interval offers a test of bound-state QED independent of the proton's structure uncertainties, it would be useful to point out (here, or in the conclusions) how would it contribute to higher order QED effects compared to the to 1S-2S spectroscopy antihydrogen.

Author response + revisions: We thank the referee for the suggestion. We have added a sentence to the conclusions to clarify that the Sternheim interval cancels out leading nuclear effects that affect both the ground state hyperfine splitting frequency as well as the 1S – 2S transition frequency.

Page 3

- Figure 1 labels: half of the fonts are broken. In any PDF viewer.

Author response + Revisions: Our apologies for the formatting/font issue that seems to have appeared at some stage of the process. This will be corrected in the final version.

Page 5

- Line 16 - What does the assumption entail that the “Breit-Rabi diagram for ground-state antihydrogen is analogous in structure to that of hydrogen”? Are you assuming the exact same slope in the fit, or only the linear approximation in this field region? (Explain here or in methods)

Author response: Our assumption is that the ground-state energy levels for antihydrogen have the same functional form, but not necessarily the same values, as described by the Breit-Rabi formula for ground-state hydrogen. This assumption means that the magnetic field

dependence cancels for any B-field when taking the difference ($a_{1S}/h = f_{da}(B) - f_{cb}(B)$). This is true even outside the region where the individual transition frequencies are almost purely linear (which we are in). Additionally, the slopes of the energy levels/transition frequencies do not need to be the exact same as in hydrogen. Another way we could state this is that we assume that the ground state energy levels of antihydrogen have the same property of hydrogen that $a_{1S}/h = f_{da}(B) - f_{cb}(B)$.

Revisions: We thank the referee for the suggestion and have expanded on this assumption in Methods.

- Line 20 - The experiments were carried out using two different magnetic fields, with ca. 4% change in the magnetic field minimum. When changing the field, a drift attributed to flux redistribution was observed. The authors were waiting 1.5 h before starting the measurement cycle to ensure a linear drift - (74 kHz/h)? I assume this is a fit that comes from multiple ECR measurements (confirming the reproducibility in the field cycling methods), but more details are needed in Methods (See also my comment later)

Author response: The “roughly 74 kHz per hour” quoted in the paper is derived from the fits that come from the two hyperfine splitting experiments themselves (at 1.03 T and 1.07 T). This is the average of the two drifts that are fit to the positron spin resonance (PSR) frequencies as done in Fig. 4. This approximate slope as well as the 1.5 hr delay were informed by previous measurements monitoring both PSR and ECR frequencies following magnet ramps. The question of ECR measurements and reproducibility comes up again in the referee comments below and we delay our full response to those as well as revisions until that point.

- Line 40 - After c->b spectroscopy (Phase 1), the remaining c-state antihydrogen is removed (Phase 2) in order to “decrease the background” before d->a spectroscopy. After the d->a measurement (Phase 3) the remaining d-states are also removed (Phase 4). It is not mentioned in the main text whether the annihilations measured during these removal phases (Phases 2, 4) are used for the analysis (which I believe is the case, to measure the state populations), so it is unclear why Phase 4 is needed. Please add more clarity to the description of these experimental sequences.

Author response: Each measurement consists of eight replicates, with the magnets kept on for the entire period of the measurement. In each replicate, antihydrogen is accumulated, and the four phases of microwave injection are carried out. Without Phase 4 there would be d-state atoms remaining in the magnetic trap when the next replicate starts. This would create an asymmetry in the populations of c and d-state atoms in the trap that would increase as the

experiment progresses. So we use Phase 4 to ensure each replicate proceeds under identical conditions.

The annihilation counts observed during Phase 2 and Phase 4 are not directly used in the analysis. However, they are used to cross-check that the c-state population has been largely removed before Phase 3, and that the total c-state and d-state populations are roughly the same, thus ruling out ongoing issues, e.g., with vacuum. The normalizations of our model lineshapes are free parameters of the fit, as mentioned in the Data analysis section of Methods.

Revisions: We have added a sentence to “The spectroscopy experiment” clarifying the purpose of Phase 4.

Page 6

- Figure 3 - In the Y axis, the label ‘Events’ are not explained, (I assume these are normalised events to the populations in c and d states? Something more complicated?).

Author response: Indeed, that label is not explained well enough in the paper. Here ‘Events’ are the annihilation counts/events observed in Phase 1 (c states) and Phase 3 (d states) of a representative repetition.

Revisions: We have added some clarifying language to the main text in the production and trapping section (and referring to Methods) as well as more a more explicit description in the caption of figure 3.

Page 7

- (Or in Methods). A discussion about the possible differences in the empirical fits your be interesting to understand for the two frequencies. For instance, it is not clear to me how this larger width of the empirical fit for d->a sweep and c->b sweep is characterised (See my comments to Methods). In essence, this part of the Main text is hard to read without the Methods, and the methods on this part are unclear as well. I suggest extending the information on this function fit, and have a better reference to in Main.

Author response: This comment is connected to another comment on Page 12 of the text and we defer our response to deal with both in one reply.

- Figure 4 (and explaining text): the drifting magnetic field has a huge effect on the measured frequencies (1 GHz on this whole scale) while the determined uncertainty for a1S/h is on the kHz level. In my understanding (Methods), this drift was characterised in the axial direction using ECR methods, but no plot or fit is accompanying this (See my comments to Methods). I suggest to extend Methods with this information, and refer to in Main.

Author response: We believe the referee misread the scale in shown in Figure 4. As shown in figure 4 and discussed in the methods the drifting magnetic field only changes the measured frequencies by 900 kHz not by 1 GHz.

Additionally, we must emphasize that the ECR magnetic field measurements were only used to study magnetic fields in preparation for these measurements and as a complementary check of the observed drifts. They were not, however, used in the analysis in any capacity. The magnetic field drift information comes entirely from the measurement of the positron spin resonance frequencies themselves, and we use the best-fit model shown in Figure 4 to characterize this drift.

Revisions: To address this comment as well as the later one relating to Page 9 of Methods we have added an extended data figure (Extended Data Fig. 2) that shows the on-axis magnetic field drift measured using the ECR technique. We have also added a paragraph to the trap magnetic field characterization section of Methods that describes the ECR measurements we performed as a complementary monitor of the magnetic field drift during each experiment. Here we include the comparison of the magnetic field drift that is extracted from the positron spin resonance onset frequencies with that from ECR. In this paragraph we also add emphasis that these measurements were NOT used in the analysis determining the hyperfine splitting. We feel this provides the reader with additional information on the magnetic field drift in each of the experiments and clarifying how ECR was used.

Page 8

- Line 9 - It is left somewhat vague how the measurement will have an impact on the mentioned nuclear structure effects: the expression “exposes the regime” to nuclear effects should be explained here, (in terms of the Zemach radius? It is indeed on the ~50 kHz level - or some other models considered?).

Author response + revisions: Indeed, here we are primarily referring to the Zemach correction and not considering alternative models. The purpose of this statement was to point out that our measurements are now in the regime where they are sensitive to nuclear structure effects. We have added a clarification that this primarily the Zemach correction to this sentence.

As of the Sternheim interval, a more detailed description on the impact of this measurement in comparison with (the much more precise) hydrogen Sternheim interval measurements would be very welcome, also in light of 1S-2S spectroscopy of both atoms.

Author response + revisions: We thank the referee for the suggestion. In addition to the earlier mentioned changes around the Sternheim interval we have added another sentence to clarify that future 1S – 2S measurements in antihydrogen can improve our determination of a_2s and subsequently significantly improve the Sternheim interval in antihydrogen. We feel that further discussion of this interval in antihydrogen is beyond the scope of this paper and would distract from our main result.

Finally, a small comment on the author list/contributions: I was sad to hear that Joel Fajans passed away last year. Of course there is no doubt that his contributions were significant to the present work, but a footnote on this would be appropriate (and a mention in the author's contributions.)

Methods:

Page 9

The magnetic field characterisation is the most important part of the systematic error studies.

The shallow centrally located minimum in the axial field (instead of a maximally flattened one in the previous works) was mapped by ECR technique, and the drifts of this field during the magnet sequencing was characterised.

I assume this would entail a detailed study on the observed flux relaxation and the linear drift phase in which the experiment was carried out - which is extremely important for the analysis. How reproducible is the slope? It is mentioned that it 'depends a lot' on the history of the magnet, which is indeed expected. Hence, the same sequencing in the magnet was used for all of the measurements.

Please show in a figure a set of these drift plots, during multiple typical experimental sequences, and the fit on the linear drift in the field. This would be pivotal here to show the reproducibility of the B field sequence, as the gradient is used in the analysis.

Author response: We thank the referee for the suggestion. Based on this comment as well as other comments referencing ECR it is possible that the referee is assuming ECR measurements of the drift are being used in the analysis and extraction of the result. We would like to emphasize that this is not the case and that the magnetic drift (the gradient) is extracted from the positron spin resonance transitions themselves (not separate ECR measurements) by performing a linear fit of the onset parameters using two straight lines with a common slope (where the slope is a free parameter of the analysis for each series). This is a key feature of our measurement and this is described in the analysis and results section of the main text and shown in figure 4 as well as discussion in Methods.

We obtain drifts of the **PSR frequencies** of (-75.64 ± 0.05) kHz/h for the 1.07 T series and (-72.82 ± 0.04) kHz/h for the 1.03 T series. From interleaved ECR measurements between each replicate we extract a drift in the **ECR frequencies** of (-73 ± 3) kHz/h for 1.07 T and (-71 ± 3) kHz/h for the 1.03 T series. In a separate ECR only experiment, we followed the magnetic drift at 1.03 T with repeated ECR measurements following the same magnet ramp procedure and measured a drift of ECR frequency to be (-71 ± 1) kHz. The drifts of the ECR frequencies are consistent with each other as well as with the drifts measured by the PSR frequencies to within the ECR uncertainty.

To summarize, we observe consistency between ECR measurements of the magnetic field drift and the PSR frequencies measured during the actual experiments. We also observe a reproducible slope when the same magnet ramp procedure is followed. As discussed in the paper we only energize the magnets once at the start of each of the two experiments. Our technique does not require that the magnetic field drift is perfectly reproducible from ramp to ramp because we monitor the drift in each experiment through the positron spin resonance transition frequencies themselves. Our measurement only relies on the assumption that the magnetic field decay region is linear (and deviations from linearity form the B-drift uncertainty).

Revisions: We have added an extended data figure (Extended Data Fig. 2) that shows the on-axis magnetic field drift measured using the ECR technique. We have also added a paragraph to the trap magnetic field characterization section of Methods that describes the ECR measurements we performed as a complementary monitor of the magnetic field drift during each experiment. Here we include the comparison of the magnetic field drift that is extracted from the positron spin resonance onset frequencies with that from ECR. In this paragraph we also add emphasis that these measurements were NOT used in the analysis determining the hyperfine splitting. We feel this provides the reader with additional information on the magnetic field drift in each of the experiments and clarifying how ECR was used.

Page 12

A bit unclear to me how the width of the two (c->b and d->a) base functions were determined. If I understood correctly, $k=2$ and $\sigma=10$ kHz is the same for both

transitions, I assume (from your MC), but (line 16-17) says " The parameters controlling the lengths of the rising and falling edges of the empirical base function were allowed to differ". How were the residual spin-flip rates determined for these parameters? I'm assuming here that you use the 'cleanup' cycles of the remaining c and d states to do this, but I don't see it mentioned here.

Author response: Some key parameters (for the base function g and for the resolution function R) were indeed fixed and assumed to be the same for both the $|c\rangle \rightarrow |b\rangle$ and $|d\rangle \rightarrow |a\rangle$ transitions in the nominal fit while others were as fit parameters. The parameter k (of the g function), which describes the degree of the polynomial rise portion of the base function, and the broadening parameter σ_B (of the one-sided resolution function R) have been fixed as pointed out by the referee. However, the parameter σ (with no subscript) and σ_r , which are the length of the rising and falling edges of the base function, respectively, are fit parameters that are allowed to differ between the $|c\rangle \rightarrow |b\rangle$ and $|d\rangle \rightarrow |a\rangle$ lineshapes. These two parameters largely determine the "width" of the base function. These parameters were left free to accommodate for differences in spin-flip rates between the $|c\rangle \rightarrow |b\rangle$ and $|d\rangle \rightarrow |a\rangle$ transitions. Also, the n parameter, describing the power-law right-tail, was free to vary. The width of the full empirical function also includes a contribution from the fixed σ_B term in the resolution function.

The annihilation counts from the "cleanup" cycles (presumably Phase 2 and Phase 4) of each replicate are not used in the determination of the g and R functions for the empirical fit. While these windows do tell us something about spin-flip rates, the annihilations that are being driven in those windows are far off-resonance at high magnetic fields and do not contain useful information for the expected annihilation rates near the magnetic minimum observed during Phases 1 and 3.

In short, the "widths" are largely given by σ and σ_r , which are fit parameters. These fit parameters are common to all replicates (ie the width does not change from replicate to replicate) but differ between the $|c\rangle \rightarrow |b\rangle$ and $|d\rangle \rightarrow |a\rangle$ lineshapes.

Revisions: We have attempted to use clearer language in the Data analysis section of Methods to clarify how the fit works. In particular that the $|c\rangle \rightarrow |b\rangle$ and $|d\rangle \rightarrow |a\rangle$ transitions are fit separately, that k and σ_B are fixed values for both transitions, that n, σ, σ_r are fit parameters and since they are fit separately to each transition can differ, and finally added two sentences that uses Fig. 3 as an example to connect those parameters to the spin-flip rates and the widths seen in Fig. 3.

(G) References

The manuscript is generally well referenced, however I find that if historic references are used for the hydrogen atom, (questionable if all is necessary) one is not allowed to

miss the most important modern precision reference, the hydrogen 1S-2S measurement (C. G. Parthey et al., Physical Review Letters (Vol. 107, 2011) At least this should be added in the appropriate place (see point F). A general revision of the relevance of some of these references would be advised.

Author response + revisions: We again thank the referee for pointing out this oversight on our part. We have removed one of the references for the hydrogen atom and added the important reference to C. G. Parthey et al., Physical Review Letters (Vol. 107, 2011) in the abstract.

(H) Clarity and context

The abstract and introduction are appropriate and lucid, and the conclusions appropriately involve the context of other research groups. The conclusions miss some more factual details (as explained in Section F), and the style of the writing could be greatly improved in this last part.

Author response + revisions: We thank the referee for their suggestions and feedback on the conclusions section and have made improvements based on the suggestions including filling in some details.